# A Review on Mycobacteriophages: From Classification to Applications

**DOI:** 10.3390/pathogens11070777

**Published:** 2022-07-07

**Authors:** Sepideh Hosseiniporgham, Leonardo A. Sechi

**Affiliations:** 1Department of Biomedical Sciences, University of Sassari, 07100 Sassari, Italy; 2Microbiology and Virology, Azienda Ospedaliera Universitaria (AOU) Sassari, 07100 Sassari, Italy

**Keywords:** mycobacteriophage, mycobacterium, tuberculosis, paratuberculosis, NTM, theranostics

## Abstract

Mycobacterial infections are a group of life-threatening conditions triggered by fast- or slow-growing mycobacteria. Some mycobacteria, such as *Mycobacterium tuberculosis,* promote the deaths of millions of lives throughout the world annually. The control of mycobacterial infections is influenced by the challenges faced in the diagnosis of these bacteria and the capability of these pathogens to develop resistance against common antibiotics. Detection of mycobacterial infections is always demanding due to the intracellular nature of these pathogens that, along with the lipid-enriched structure of the cell wall, complicates the access to the internal contents of mycobacterial cells. Moreover, recent studies depicted that more than 20% of *M. tuberculosis* (Mtb) infections are multi-drug resistant (MDR), and only 50% of positive MDR-Mtb cases are responsive to standard treatments. Similarly, the susceptibility of nontuberculosis mycobacteria (NTM) to first-line tuberculosis antibiotics has also declined in recent years. Exploiting mycobacteriophages as viruses that infect mycobacteria has significantly accelerated the diagnosis and treatment of mycobacterial infections. This is because mycobacteriophages, regardless of their cycle type (temperate/lytic), can tackle barriers in the mycobacterial cell wall and make the infected bacteria replicate phage DNA along with their DNA. Although the infectivity of the majority of discovered mycobacteriophages has been evaluated in non-pathogenic *M. smegmatis*, more research is still ongoing to find mycobacteriophages specific to pathogenic mycobacteria, such as phage DS6A, which has been shown to be able to infect members of the *M. tuberculosis* complex. Accordingly, this review aimed to introduce some potential mycobacteriophages in the research, specifically those that are infective to the three troublesome mycobacteria, *M. tuberculosis*, *M. avium* subsp*. paratuberculosis* (MAP), and *M. abscessus,* highlighting their theranostic applications in medicine.

## 1. Introduction

Mycobacteriophages are tailed double-stranded DNA (dsDNA) viruses belonging to the *Caudovirales* order, which includes two principal families, *Siphoviridae* and *Myoviridae*. Members of these two families are distinguished based on their morphological and evolutional divergences, in which they are distributed into 29 clusters [1] and 10 singletons [2] to date. The majority of discovered mycobacteriophages have been isolated using *M. smegmatis* mc^2^155 [3]. Some others are only known to infect *M. tuberculosis* H37Rv [3,4], while others have been isolated using other mycobacterial species such as vB_MapS_FF47 [5] that can infect MAP ATCC19698 [5]. The abilities of mycobacteriophages in crossing the waxy structure of the cell wall [6] and infecting mycobacteria have made them potential tools in the diagnosis and treatment of infections caused by pathogenic mycobacteria [7,8]. Mycobacterial diseases (tuberculosis or non-tuberculous [9]) are categorized among the most demanding conditions that jeopardize the lives of many humans, specifically patients who are suffering from immune-mediated inflammatory diseases [9] and sick animals [10] annually. The emergence of multi-drug resistant (MDR) mycobacteria has raised many questions about the efficiency of antibiotics. MDR in *M. tuberculosis* is defined as the tolerance of the bacterium to the antibiotics Isoniazid and Rifampicin [11]. A survey between 2009 and 2016 showed that the rate of MDR in patients suffering from *M. tuberculosis* infections increased by 20% [12]. This is under the condition that the susceptibility of non-tuberculosis mycobacteria to first-line tuberculosis antibiotics has also declined in recent years [13]. Mycobacteria can acquire antibiotic resistance genes intrinsically or extrinsically. Studies demonstrated that *M. tuberculosis* is inherently resistant to many antibiotics, which can undermine the functionality of many drugs against the pathogen [14]. The intrinsic antibiotic tolerance in mycobacteria comes from the thick-waxy-hydrophobic structure of the cell wall and the presence of mycobacterial enzymes that can modify or degrade drugs [14]. Chromosomal mutations contribute a possible role in antibiotic resistance of *M. tuberculosis*. These mutations could promote the overexpression of drug targets and inhibit prodrug activation [14].

Today, mycobacteriophages in isolation/cocktail [15] or within a synergistic therapy along with antibiotics opened new horizons in the treatment of infections caused by pathogenic mycobacteria [16,17,18]. Furthermore, mycobacteriophages have been exploited in diagnostic approaches of slow-/fast-growing mycobacteria through shuttle plasmids that express resistance against antibiotics and induce the production of lysogenic plaques in target mycobacteria [19]; transduction of nanoluciferase reporter gene cassettes that discriminate viable drug-resistant and -sensitive strains [20]; phage amplification for detection of antimicrobial resistance species [21,22,23]; and viability assessment of target mycobacteria in various samples (i.e., dairy products, blood, stool [24,25]). This review aimed to look back on some of the most used mycobacteriophages in research, elucidating their applications in the diagnosis and treatment of drug-resistant pathogenic mycobacteria with a focus on three pathogenic mycobacterial species, *M. tuberculosis*, MAP, and *M. abscessus,* from past to now.

## 2. An Explanation of Discovered Mycobacteriophages Infective to *M. smegmatis*, *M. tuberculosis*, *M. bovis, M. avium* Subspecies, and *M. abscessus*

This part of the review lists some of the most interesting mycobacteriophages, which might be used in future diagnostic and therapeutic studies against three pathogenic mycobacteria of *M. tuberculosis*, *M. avium* subsp*. paratuberculosis*, and *M. abscessus,* highlighting their properties and potentials based on previous studies (Table 1).

Many mycobacteriophages have been discovered that are primarily screened and replicated in fast-growing mycobacteria, specifically *M. smegmatis* mc^2^155 [5]. Some of the mycobacteriophages that were isolated using *M. smegmatis* mc^2^155 are as follows: C2, I3, Bxz1, Rahel, D4, D29, L1, L5, PDRPxv, OKCentral2016, Ms6 Donny, BPs, Angel, Halo, Faze9*,* Donny, KingMidas, ZoeJ, Milly, Adephagia and CrimD, and phAE19.

Generalized transduction has been reported in several mycobacteriophages that existed in cluster C, such as I3 and Bxz1 [26,27]. Members of this cluster are distinguished by their capabilities in generalized transduction of genetic markers in *M. smegmatis* [28], a fast-growing mycobacterium, but not in slow-growing mycobacteria such as *M. tuberculosis* [28]. Generalized transduction is a common phenomenon among bacteriophages, characterized by the transition of gene segments from a donor bacterium to a recipient one [29,30]. Therefore, generalized transduction would be a potential area in future diagnostic and therapeutic studies that needs to be inspected with various mycobacteriophages such as Rahel, another member of cluster C.

D4 and D29 are two mycobacteriophages that can not only replicate in *M. smegmatis* but also become inactivated by the lipid extract of *M. smegmatis* in an appropriate incubation time. This is due to the adsorption phenomenon [31,32]. Some studies reported that peptidoglycolipids or general mycoside C on the surface of some mycobacterial species and glycolipids on the cell wall of *M. phlei* [33] act like receptors that adsorb and inactivate mycobacteriophages [33,34,35,36,37]. This inactivation would be accompanied by the adsorption of the phage to the superficial phage-specific receptors on the surface of mycobacteria [34,35,36,37]. L1 [38,39] and L5 [40,41,42] are mycobacteriophages that can induce the formation of superinfection-stable lysogens [38,39] and, meanwhile, have toxicity against *M. smegmatis*. Both mycobacteriophages could infect fast- and slow-growing mycobacterial species with the difference that the calcium concentration is critical in L5 infectivity [43,44]. Gp63 and gp64 genes in L5 and hlg1 (the identical ortholog of L5 gp64) in L1 take part in toxicity in these mycobacteriophages [45]. Genomic analysis revealed that the stability of lysogens against L5 superinfection is guaranteed by immunity acquired by a peptide of 183 amino acids encoded by gene 71 in L5. This peptide acts as a phage repressor and does not permit any other phages to re-infect lysogens by prohibiting the lytic cycles in superinfecting homo-immune viruses [38,46]. It has been suggested that the capability of mutant mycobacteriophage L1 in the production of higher-yielding progenies could be exploited in the diagnosis of mycobacteria [39], in which an analysis of a temperature-sensitive mutant mycobacteriophage L1 (L1-P2) depicted that frequent transduction of this mycobacteriophage could increase the yield and efficiency of the phage inversely [39].

Mycobacteriophages encode endolysins (lytic enzymes) that can promote the lysis of phage-infected bacteria at the end of propagation [47]. Mycobacteriophages are distributed into three categories based on their capability to produce one or two endolysins, including Lysin A, Lysin A and Lysin B, and Lysin B [48]. Lysin A and Lysin B can disintegrate peptidoglycan and mycolic acid-arabinogalactan layers, respectively [48]. All bacteriophages have lysin A domains in their genomes, whereas Lysin B is specific to some mycobacteriophages [49]. A study on mycobacteriophage Che12 revealed that Gp11 in this phage has Chitinase domains that function as Lysin A. Che12 lysin A cuts off NAG-NAM-NAG of the peptidoglycan structure in the mycobacterial cell wall and produces tautomers of NAG-NAM-NAG [50]. Genomes in mycobacteriophages PDRPxv and OKCentral2016 [51] can encode the two endolysins [48], whereas mycobacteriophages Ms6 [4], Donny [52], and D29 only have genes that induce the production of Lysin B [51,53]. Endolysin extracted from PDRPxv could affect both dividing and non-dividing host cells. Mycobacterial endolysins have potential therapeutic properties against drug-resistant tuberculosis [48]. The turbidimetric and biochemical analyses revealed that PDRPxv-recombinant endolysins have antimicrobial impacts on *M. smegmatis* [48]. Lysin A has a secretory role and is expressed in periplasmic spaces to deteriorate its target, peptidoglycan. This explains that PDRPxv could lyse the bacterial hosts without the need for transmembrane holin protein [48]. The question that might arise here is how endolysin can penetrate and disrupt the mycobacterial cell wall without holin. A holin-independent lysis strategy has been predicted in PDRPxv that assists the secretion of endolysins into the cell wall via either secretory signals presented in endolysins [54] or chaperone proteins other than holin [55]. Additionally, the result of Western blot analysis of Lysin A revealed that this endolysin is expressed in the periplasmic area, which is why Lysin A could reach peptidoglycan and destroy it in the absence of holin [48].

OKCentral2016 has remarkable proximity to some members of subcluster A10, such as Goose and Twister, in terms of possession of gene sequences that encode Lysin A, Lysin B, terminase, portal, capsid maturation protease, scaffolding, and major capsid subunit proteins [51]. This is under the condition that Ms6 LysB has an appendix in the N-terminal (N-*Terminus)* that has an affinity to the peptidoglycan-binding domain (PGBD) of φKZ endolysin. Later, the functionality of LysBPGBD was inspected by incorporation of the region with an enhanced green fluorescent protein (EGFP). The result indicated that the pack of LysBPGBD-EGFP could successfully attach to *M. smegmatis*, *M. vaccae*, *M. bovis* BGC, and *M. tuberculosis* H37Ra cells pretreated with SDS or Ms6 LysB [53]. The result of a study in 2019 depicted that Lysin B induced by phage D29 has antimicrobial profiles against *M. ulcerans,* in which recombinant lysin B could induce the lytic cycle in *M. ulcerans* isolates [56].

Similar to mycobacteriophage L5, the integration of Ms6 into the mycobacterial genome could impose no changes on the phage genome. Ms6 was primarily used as a recombinant vector for transducing the aph gene of the Tn5 transposon into the genome of *M. smegmatis*, in which the aph gene in the lysogen of *M. smegmatis* remained intact even after 150 generations in the absence of Kanamycin (to isolate the transformed mycobacteria resistant to Kanamycin, Myco agar, including 4.7 g Middlebrook 7H9, 5 g nutrient broth, 10 mL 50% glycerol, 0.05% Tween 80 per liter, and 1.2% agar supplemented with 20 μg/mL Kanamycin [4]). This attribute of Ms6 makes it a potential selective carrier for the transduction of foreign DNA into the genome of *M. smegmatis* [4].

Interestingly, three mycobacteriophages of BPs, Angel, and Halo have the smallest genome size among other mycobacteriophages with 41,901 bp, 42,289 bp, and 41,441 bp, respectively [57]. Analysis depicted that all three phages could infect *M. tuberculosis.* However, BPs and Halo replicated in *M. tuberculosis,* producing fewer plaques on an original plate containing an *M. tuberculosis* lawn, and this is because they could not efficiently identify their receptors on *M. tuberculosis* cells [57]. Further studies depicted that Halo’s capability in forming plaques on a lawn of *M. tuberculosis* has increased by replating a single plaque taken from the primary *M. tuberculosis* lawn. This suggests that replating Halo in a new *M. tuberculosis* lawn would induce mutations to the phage genome, and this may significantly enhance the chance of competent attachment of the phage to *M. tuberculosis* receptors. Despite the importance of all three phages, more studies have been focused on BPs and Angel due to the presence of insertions of two extremely small mobile genetic elements, MPME1 and MPME2, respectively. Studies depicted that the presence of tRNA in mycobacteriophages highly depends on the cluster type. Although members of clusters C, L, M, and V possess multiple tRNAs, tRNA does not exist in cluster B at all [58]. Clusters A, E, and K were detected with either no tRNA or a minimum of one tRNA [58]. tRNA genes might enable mycobacteriophages to either replicate in their host cells or infect a broader range of mycobacteria [58]. Accordingly, this could be interpreted that mycobacteriophages with tRNA could better propagate in their host cells rather than phages without tRNA. A study on a genetically manipulated mycobacteriophage T4 that lost its tRNA genes revealed that mutant phages produced less protein and smaller plaques than their wild type [58]. On the other hand, it was suggested that mycobacteriophages that possess a few tRNAs are probably temperate phages that underwent recombination and obtained the corresponding DNA from their bacterial hosts [58]. Some temperate mycobacteriophages can protect lysogeny by hindering the penetration of other superinfecting mycobacteriophages and evading other prophage defenses. Superinfection immunity has been reported among members of cluster K, such as mycobacteriophages ZoeJ and Milly from subcluster K2, Adephagia and CrimD from subcluster K1. ZoeJ is a novel temperate phage that can infect slow- and fast-growing mycobacteria such as *M. smegmatis*, *M. tuberculosis*, and MAP. Significant proximity was discovered between ZoeJ and TM4 (both for cluster K). However, a precise deletion of repressor and integrase genes led to the dominance of lytic phenotype and prevention of superinfection immunity in TM4 (e.g., formation of clear plaques) [59]. Gene 45 (gP45) in ZoeJ plays an important role in superinfection immunity and lysogeny. This gene is encoded by prophages in Milly and Adephagia phages, while it is deleted in other mycobacteriophages such as TM4 [59].

FRAT1 is a temperate mycobacteriophage that induces the lysogenic cycle in both *M. smegmatis* and *M. bovis* (BCG). FRAT1 would be a potential tool for detecting antibiotic resistance among *M. bovis* (BCG) isolates. In 1992, a study on the genome of FRAT1 depicted that an 11-Kbp fragment of phage DNA belonging to the *ClaI* gene was excised by the integration of FRAT1 into the *M. smegmatis* genome [60]. Further analysis revealed that the deleted fragment belonged to the Kanamycin resistance gene, in which the exposure of a set of PJRD184 (plasmid) to *M. smegmatis* cells that were infected with FRAT1 led to the transformation of plasmids. Accordingly, a Kanamycin-resistant gene (TN903) that overlapped with the deleted 11-Kbp *ClaI* gene appeared in plasmids [60].

The discovery of mycobacteriophages that could replicate in *M. tuberculosis* strains was a significant improvement, raising hopes about speedy detection and treatment of *M. tuberculosis* infections. Previous studies revealed that mycobacteriophages belonging to subcluster A2, A3, K1, and K4 are more likely to infect *M. tuberculosis* strains [61]. Several mycobacteriophages were discovered that could propagate in either *M. tuberculosis*, such as Ds6A, or other mycobacterial species (i.e., *M. smegmatis* mc^2^155), such as mycobacteriophages D28, D29, D32, L5, Bo4, 33D, SWU1, TM4, ZoeJ, and Che12. Engineering shuttle plasmid is one of the diagnostic applications of mycobacteriophages that could infect *M. tuberculosis* strains. DS6A and TM4 have been used for the generation of shuttle plasmids in several studies [62]. In a comparative study in 2016, the DS6A fluorophage shuttle plasmid was produced using DS6A, and its infectivity was assessed against members of the *M. tuberculosis* complex (MTBC), such as *M. tuberculosis,* and non-tuberculosis mycobacteria (NTMB), such as *M. avium* and *M.*
*fortuitum* [63]. To generate this fluorophage, specific regions that had no essential functionality in a highly intact phage genome were removed and then replaced with dephosphorylated plasmid fragments. The result of this study depicted that DS6A only formed plaques on MTBC strains [63]. Assessment of DS6A infectivity via flow cytometry analysis of other non-tuberculosis mycobacteria (NTMB) demonstrated that DS6A infected most of the studied mycobacteria. This suggests that plaque formation is not equal to phage infection and fluorescence emission. Productive infection via DS6A is defined as the capability of the phage to complete the cycle of infection in host cells, including (1) initial step: adsorption, attachment, and DNA injection; (2) phage amplification; and (3) plaque formation [63]. The capability of DS6A in infecting specific MTBC hosts could be supported by phylogenetic analysis demonstrating that the DS6A genome has a unique mosaic structure [63]. Interestingly, a comparative study on the impact of exposure of *M. tuberculosis* strain H37Rv to DS6A and GS7 demonstrated that DS6A could deteriorate the acid-fast characteristic of *M. tuberculosis* since it could consequently infect and lyse the bacterium, whereas GS7 had no impact on the acid-fastness of the bacterium because it could not lyse *M. tuberculosis* [64]. The specificity of DS6A in interacting with limited numbers of mycobacteria made it an ideal phage in treatment [65,66] and diagnosis [67,68,69,70] of MTBC-relevant infections [63].

On the other hand, the TM4 phage can replicate in both slow- and fast-growing mycobacteria, such as *M. smegmatis* and *M. tuberculosis* or *M. avium* or *M. paratuberculosis,* respectively [62,71]. This mycobacteriophage has always been a perfect model for creating recombinant shuttle vectors within diagnostic studies where it could transduce reporters and transposons to mycobacterial genomes [62]. TM4 belongs to subcluster K2 [59,62,72]. In contrast to other members of cluster K that can form stable lysogens and turbid plaques, TM4 creates hazy [62] to clear plaques in solid media [72,73]. TM4 could induce the formation of pseudolysogens in some mycobacteria and the production of proteins that have similar structures to haloperoxidases, glutaredoxins, and the WhiB family of transcriptional regulators [62]. Some studies suggested that TM4, L5, and D29 might have a common ancestor since several similar DNA fragments were found in TM4, L5, and D29 that express similar proteins in these phages [62].

The ability of some mycobacteriophages in lysing pathogenic mycobacteria has also been examined in previous studies. Bo4 and 33D are mycobacteriophages that can induce the lytic cycle in host cells. Bo4 lyses mycobacteria such as *M. smegmatis* and specifically *M. tuberculosis* in various environments such as bloodstream and lysosomal macrophages with a pH of 5 or 7.4 [74]. This feature, plus the issue that Bo4 has no destructive content or gene that could either enhance mycobacterial virulence or deteriorate the human immune system, made Bo4 a selective tool for detecting and treating *M. tuberculosis* infections [74]. In 1979, 33D (Warsaw) was primarily used for therapeutic purposes [75]. Since 33D could lyse the most troublesome mycobacteria close to *M. bovis* strain Bacillus Calmette-Guerin (BCG), such as *M. bovis* and *M. tuberculosis,* but not BCG; this discriminative capability might be exploited in the detection and isolation of BCG where it is considered a potential treatment for tumors [75]. A primary study on the isolation of CRB2 depicted that it could effectively infect both *M. smegmatis* and *M. tuberculosis* [76]. Experiments for finding the ability of CRB2 in transducing chromosomal genes indicated that CRB2 is a lytic non-transducing mycobacteriophage. Moreover, the result of pulsed-field gel electrophoresis (PFGE) demonstrated that CRB2 produced diverse lengths of chromosomes, suggesting that CRB2 has redundant ends and that the phage applies a headful-packaging strategy [77] to wrap its DNA, and this might lead to the creation of virion particles that carry host DNA [76,78].

Mycobacteriophage SWU1 was firstly isolated from soil in Sichuan in China [79]. This lytic phage can replicate in *M. smegmatis* mc^2^155 and *M. tuberculosis* [79]. SWU1 could affect phage-infected mycobacteria via various strategies, including (1) modification of cell signaling, which can influence growth, pathogenesis, and cell wall metabolism in mycobacteria; (2) modification of cell energy and ion fluxes; (3) dispossessing the replication system of phage-infected mycobacteria; (4) interference with iron uptake system [42].

Interestingly, SWU1 forms bull’s eye-like plaques on *M. smegmatis* mc^2^155 lawn, and plaques are distinguished from adjacent ones via a line-like feature. Multiple sequence alignment in DNAman and NCBI megablast comparison revealed that DNA sequences in SWU1 and L5 mycobacteriophages are highly similar by 94.66% and 97%, respectively [80]. However, they are morphologically detectable. In comparison, L5 induces the production of turbid plaques, and SWU1 forms clear-turbid-clear-turbid circles on a lawn of *M. smegmatis* mc^2^155. Additionally, phylogenic analysis depicted that SWU1 and L5 genomes underwent insertion and deletions that imposed dissimilarities on their genome structures. Che12 is a temperate mycobacteriophage that can infect and lysogenize *M. tuberculosis* [81]. The temperate profile of this phage was discovered via superinfection immunity analysis, and the integration of the phage into *M. tuberculosis* was guaranteed by a southern hybridization experiment using Che12 DNA as a probe. Che12 has a genome similarity above 80% with L5 and D29, in which phylogenic studies suggested that Che12 might be derived from L5. Interestingly, the Che12 attachment site *attP* has homology to *attB* in *M. smegmatis* and *M. tuberculosis,* and this capability could be exploited in the phage-based diagnosis of *M. tuberculosis* [81]. In 2009, a recombinant Che12, phAETRC16, was engineered to carry a luciferase cassette infecting viable *M. tuberculosis* and visualize the presence of the bacterium in specimens via emission of luciferase light in which the emitted light was measurable by a luminometer [82]. In another attempt, recombinant Che12 was produced by substituting the pYUB328 region in phAETRC10 with hygromycin resistance marker luciferase [82]. This recombinant Che12 was employed to convey the hygromycin resistance cassette into *M. tuberculosis* lysogens to evaluate the lysogenicity in *M. tuberculosis* and differentiate the resistant-to-hygromycin lysogens from sensitive ones [82].

D29, TM4, vB_MapS_FF47, and ZoeJ are some of the mycobacteriophages infective to MAP. D29 is a lytic phage [83] that has a close genomic affinity to mycobacteriophage L5 [83] and Che12 [81]. D29 has predominantly been used in the assessment of MAP viability in various sample types, such as milk [84], blood [85], tissue [86], feces [86], and direct capture of MAP prior to viability assessment in milk samples [87,88]. D29 is stable at pH between 9 to 10. A study on the assessment of infectibility of *M. smegmatis* and MAP via D29 and TM4 under oxygen restriction demonstrated that TM4 and D29 could comparably attach to MAP and *M. smegmatis* surfaces under oxygen ban. However, D29 could not proceed to the infection step, and just TM4 successfully infected the viable cells. This is under the condition that the infectivity of D29 was retrieved after incubating samples in an oxygen-enriched environment for at least one hour [89].

In 2014, a new method for making a confluent growth culture of MAP was optimized that did not need a lawn-making step via fast-growing mycobacteria (i.e., *M. smegmatis*), in which the assay sped up the recovery of MAP incubated at 30 °C for 4–6 weeks. Later, the assay was tried and conveniently isolated a dsDNA lytic mycobacteriophage from the bovine feces [5]. This mycobacteriophage contained no virulent or temperate genes and was called vB_MapS_FF47 [5]. This phage screening method would facilitate the isolation of novel mycobacteriophages, infectious to the pathogenic mycobacteria, such as MAP and *M. tuberculosis,* in the future [5]. FF47 belongs to the *Siphoviridae* family and has proximity to mycobacteriophage Muddy and *Gordonia* phage (GTE2) [5]; however, FF47 cannot replicate in *Gordonia*, *Rhodococcus*, or *Nocardia* spp., same as GTE2.

*M. abscessus* is another concerning mycobacterial species that has been noticed in phage diagnosis and therapy studies. Araucaria and phiT46-1 are mycobacteriophages that can infect *M. abscessus.* Araucaria is a temperate mycobacteriophage and was isolated from respiratory tract samples that were simultaneously contaminated with *M. abscessus* subsp. *Bolletii* [90]. The structure of capsid and connector in Araucaria are similar to Gram-positive and -negative bacteriophages; however, Araucaria has a helical tail embellished with radial appendixes [90].

Studies depicted that *M. abscessus* could produce and liberate prophages spontaneously [91,92,93,94]. phiT46-1 is one of the phages released from the *M. abscessus* strain Taiwan-46. In contrast to many mycobacteriophages, phiT46-1 cannot infect *M. smegmatis*, precisely replicating in the *M. abscessus* strain BWH-C. This mycobacteriophage contains a polymorphic toxin-immunity system that is connected to secretory systems VII [95]. Although phiT46-1 has several virion structural genes similar to cluster Q mycobacteriophages, the genomic affinity between phiT46-1 and other actinobacteriophages is estimated to be less than 4% [95]. Other studies have also confirmed the presence of prophages in some *M. tuberculosis* strains. phiRv1 (φRv1) and phiRv2 (φRv2) are two prophages recognized in *M. tuberculosis* H37Rv and CDC1551, respectively. phiRv1 is located within the REP13E12 repeated sequence and possesses components necessary for integrating the prophages into the bacterial genome, including the *attB* site and serin recombinase family (Rv1586c) segment [96]. Furthermore, the recent two prophages carry genes that contribute to the production of virus-like particles [96].

**Table 1 pathogens-11-00777-t001:** Descriptive comparison of some potential mycobacteriophages infective to *M. tuberculosis*, *M. bovis, M. avium* spp., and *M*. *abscessus*.

Mycobacteriophage/Family	Description	Cluster/Sub Cluster	Origin	CG% Content	Infect	Life Cycle	Completely Sequenced
Bxz1/*Myoviridae* [97]	Generalized transduction, Bxz1-specific tRNA [27]	C [27]	Soil [26]	64.8 [27]	*M. smegmatis* mc^2^155, *M. vaccae* [26]	Lytic (Clear plaques) [26]	Yes [27]
L5/*Siphoviridae* [97]	Superinfection-stable lysogens, transformation of slow-growing mycobacteria, immobilized tail protein (Gp6) [40,41,42], three tRNA genes [27]	A/A2 [40,41,42]	Isolated from lysogenic strain of *M. smegmatis* [98]	63.2 [98]	*M. smegmatis* mc^2^155, *M. tuberculosis* [44]	Temperate [99]	Yes [98]
PDRPv/*Siphoviridae* [100]	Antimicrobial profiles [100], circular permuted dsDNA [100]	B/B1	Soil	66 [100]	*M. smegmatis* mc^2^155*, M. tuberculosis*	Lytic [100]	No
D29/ *Siphoviridae* [49]	Lytic activity, inactivation by *M. smegmatis* extracted mycoside C [34,35,36,37], adsorption [31,32], Lysin B [56]	A/A2 [40]	Soil [101]	63.6 [102]	*M. smegmatis* mc^2^155*, M. tuberculosis* [83], MAP, *M. bovis*, *M. fortitum* [88]	Lytic [83]	Yes [83]
BPs/*Siphoviridae* [57]	Ultra-small genetic elements	G [57,103]	Soil	66.6 [57]	*M. smegmatis* mc^2^155 [57], *M. tuberculosis* [57]	Temperate [57]	Yes [57]
Angel/*Siphoviridae* [57]	Ultra-small genetic elements	G [57,103]	Soil	66.6 [57]	*M. smegmatis* mc^2^155 [57], *M. tuberculosis* [57]	Temperate [57]	Yes [57]
Halo/*Siphoviridae* [57]	Ultra-small genetic elements	G [57,103]	Soil	66.7 [61]	*M. smegmatis* mc^2^155 [57], *M. tuberculosis* [57]	Temperate [57]	Yes [57]
ZoeJ/*Siphoviridae* [59]	Superinfection immunity [59]	K/K2 [59]	Soil [104]	Unpublished	*M. smegmatis* mc^2^155*,* *M. tuberculosis, M. avium, M. bovis* [59]	Temperate [59]	Yes [59]
TM4/*Siphoviridae* [73]	Genetic tools [62,72], diagnostic application, unusual lysogenic pattern, production of proteins similar to transcriptional regulators, generation shuttle plasmid [62]	K/K2 [59]	Unknown	68.1 [27]	*M. smegmatis* mc^2^155 [62]*, M. tuberculosis* H37Rv [59,62]*, M. avium*, MAP	Temperate	Yes [27]
FRAT1/Unknown	Integrase gene [105]; carries Kanamycin resistance gene; therapeutic tools [60]	Unknown	Unknown	Unknown	*M. smegmatis* ATCC607, *M. bovis* BCG 1173/P2 [60]	Temperate [105]	No
D32/*Siphoviridae* [106]	Lytic activity against *M. tuberculosis*	Unpublished	Soil	64 [107]	*M. tuberculosis* H37Rv [108], *M. smegmatis ATCC607, M. smegmatis* mc^2^155 [106]	Lytic [101]	Yes [106]
Bo4/*Siphoviridae* [109]	Lytic activity, active in bloodstream and lysosomal macrophages [74]	G [74]	Unknown	66.76 [74]	*M. smegmatis* CMCC93202, *M. tuberculosis* H37Rv [74]	Lytic [74]	Yes [74]
33D/*Siphoviridae* [109]	Lytic activity, therapeutic purposes [75]	Unknown	Unknown		*M. tuberculosis* H37Rv and *M. bovis* (TMC410) [75]	Lytic [75]	No
SWU1/*Siphoviridae* [109]	Lytic activity, modification of cell signaling, bull’s eye morphology [42]	A2 [42]	Soil [110]	62.4 [110]	*M. smegmatis* mc^2^155 [42], *M. tuberculosis* [79].	Lytic [80]	Yes [80]
Che12*/Siphoviridae* [111]	Diagnosis of tuberculosis [82]	A/A2 [61]	Soil [82]	62.9 [97]	*M. tuberculosis* H37Rv [112] and *M. smegmatis* mc^2^155 [82]	Temperate [82]	Yes [97]
DS6A*/Siphoviridae* [109]	Formation of plaque only on MTBC, loss of acid fastness, generation of shuttle plasmid [62]	Singleton [3]	Unknown	68.4 [63]	*M. tuberculosis* H37Rv [113], *M. tuberculosis* complex [63]	Temperate	Yes [114]
CRB2*/Siphoviridae* [76]	Non-transducing profile; ORFs in its genome have a probable function [76]	B/B9 [76]	Unknown	69.78 [76]	*M. smegmatis* mc^2^155 and *M. tuberculosis*	Lytic [76]	Yes [76]
vB_MapS_FF47*/Siphoviridae* [5]	Lytic activity, no virulent or temperate genes [5]	Unpublished	Bovine feces [5]	58.6 [5]	MAP ATCC19698 [5] and *M. smegmatis* mc^2^155 [5]	Lytic [5]	Yes [5]
AN3*/Siphoviridae* [115]	Used for typing of *M. avium intracellular scrofulaceum* [116]	Unpublished	Unknown	Unpublished	*M. smegmatis* mc^2^155 [115] and *M. avium intracellular scrofulaceum* [116]	Unpublished	Yes [115]
AN9*/Siphoviridae* [117]	Used for typing of *M. avium intracellular scrofulaceum* [116]	Unpublished	Unknown	Unpublished	*M. smegmatis* mc^2^155 [117] and *M. avium-intracellulare-scrofulaceum* complex [116,118]	Unpublished	Yes [117]
ANI8*/Siphoviridae* [119]	Used for phage typing of *M. avium intracellular scrofulaceum* (MAIS) [118]	Unpublished	Unknown	Unpublished	*M. smegmatis* mc^2^155 [119] and *M. avium intracellular scrofulaceum* (MAIS) [118]	Unpublished	Yes [119]
VA6*/Siphoviridae* [120]	Used for typing of *M. avium intracellular scrofulaceum* [116]	Unpublished	Unknown	Unpublished	*M. smegmatis* mc^2^155 [120] *and M. avium intracellular scrofulaceum* [116]	Unpublished	Yes [120]
VC3*/Siphoviridae* [121]	Used for typing of *M. avium intracellular scrofulaceum* [116]	Unpublished	Unknown	Unpublished	*M. smegmatis* mc^2^155 [121] and *M. avium intracellular scrofulaceum* [116]	Unpublished	Yes [121]
Muddy*/Siphoviridae* [122]	Lytic activity [123]	AB [123]	Soil [122]	Unpublished	*M. smegmatis* mc^2^155 [16], *M. abscessus* (GD01) [16]	Lytic [123]	Yes [122]
Araucaria/*Siphoviridae* [90]	Infection via adhesion to cell wall saccharide and protein [90]	Dori-like [90]	Respiratory tract sample [90]	64.41 [90]	*M. abscessus* subsp. *bolletii* CIP108541 [90]	Temperate [90]	Yes [90]
Prophage phiT46-1/*Siphoviridae* [95]	Polymorphic toxin-immunity cassette [95]	Unpublished	It was isolated by spontaneous release from *M. abscessus* strain Taiwan-46 [95]	64 [95]	*M. abscessus* strain BWH-C [95]	Temperate [95]	Yes [95]
Prophage phT45/*Siphoviridae* [124]	Polymorphic toxin-immunity cassette associated with type VII secretion systems [124]	Unpublished	It was isolated by spontaneous release from *M. abscessus* strain Taiwan-45 [124]	65 [124]	*M. abscessus* strain BWH-C [124]	Lytic [124]	Yes [124]
Adler [125]	Genes encoding cytochrome P450 (heme protein) catalyze monooxygenase activity [125]	Unpublished	Unknown	Unknown	*M. abscessus subspecies bolletii F1660* [125]	Unknown	No
Chancellor*/Siphoviridae* [126]	Virion structure and assembly genes, lytic activity, Lysin A, Lysin B, holin genes, ability to infect *M. tuberculosis* [126]	K/K4 [126]	Soil [126]	68 [126]	*M. smegmatis* mc^2^155 [126], predicted to infect *M. tuberculosis* [126]	Temperate [126]	Yes [126]
Mitti*/Siphoviridae* [126]	Virion structure and assembly genes, lytic activity, Lysin A, Lysin B, holin genes, ability to infect *M. tuberculosis* [126]	K/K4 [126]	Soil [126]	68 [126]	*M. smegmatis* mc^2^155 [126], predicted to infect *M. tuberculosis* [126]	Temperate [126]	Yes [126]
Wintermute*/Siphoviridae* [126]	Virion structure and assembly genes, lytic activity, Lysin A, Lysin B, holin genes, ability to infect *M. tuberculosis* [126]	K/K4 [126]	Soil [126]	68 [126]	*M. smegmatis* mc^2^155 [126], predicted to infect *M. tuberculosis* [126]	Temperate [126]	Yes [126]
ShedlockHolmes */Siphoviridae* [127]	Ability to infect *M. tuberculosis*, having tRNA [127]	K/K3 [127]	Soil [127]	67.3 [127]	*M. smegmatis* MC^2^155 [127], predicted to infect *M. tuberculosis* [127]	Temperate [127]	Yes [127]
Deby*/Siphoviridae* [128]	Ability to infect *M. tuberculosis,* structural and assembly genes*,* having tRNA [128]	K/K1 [128]	Soil [128]	66.5 [128]	*M. smegmatis* mc^2^155 [128], predicted to infect *M. tuberculosis* [128]	Temperate [128]	Yes [128]
LaterM*/Siphoviridae* [128]	Ability to infect *M. tuberculosis,* structural and assembly genes, having tRNA [128]	K/K1 [128]	Soil [128]	66.5 [128]	*M. smegmatis* mc^2^155 [128], predicted to infect *M. tuberculosis* [128]	Temperate [128]	Yes [128]
LilPharaoh*/Siphoviridae* [128]	Ability to infect *M. tuberculosis,* structural and assembly genes, having tRNA [128]	K/K1 [128]	Soil [128]	67.1 [128]	*M. smegmatis* mc^2^155 [128], predicted to infect *M. tuberculosis* [128]	Temperate [128]	Yes [128]
SgBeansprout*/Siphoviridae* [128]	Ability to infect *M. tuberculosis,* structural and assembly genes, having tRNA [128]	K/K1 [128]	Soil [128]	67.1 [128]	*M. smegmatis* mc^2^155 [128], predicted to infect *M. tuberculosis* [128]	Temperate [128]	Yes [128]
Sulley*/Siphoviridae* [128]	Ability to infect *M. tuberculosis,* structural and assembly genes, having tRNA [128]	K/K1 [128]	Soil [128]	66.4 [128]	*M. smegmatis* mc^2^155 [128], predicted to infect *M. tuberculosis* [128]	Temperate [128]	Yes [128]
Paola*/Siphoviridae* [128]	Ability to infect *M. tuberculosis,* structural and assembly genes, having tRNA [128]	K/K5 [128]	Soil [128]	65 [128]	*M. smegmatis* mc^2^155, predicted to infect *M. tuberculosis* [128]	Temperate [128]	Yes [128]
Joy99*/Siphoviridae* [129]	Ability to infect *M. tuberculosis;* its genome contains genes that contribute in virion assembly/structure/lysis proteins/host integration and excision proteins/DNA primase/RusA-like resolvase/RtcB-like integrase genes; having tRNA [129]	K/K1 [129]	Soil [129]	66.6 [129]	*M. smegmatis* mc^2^155 [129], predicted to infect *M. tuberculosis* [129]	Unpublished, three-ring morphology with clear center spot, thin middle ring, and turbid outer ring [129]	Yes [129]
20ES*/Siphoviridae* [130]	Capability to infect *M. tuberculosis,* presence of *Par*A and *Par*B genes in its genome [77]	A [77]	Soil [130]	63.43 [77]	*M. smegmatis* mc^2^155 [130], able to infect *M. tuberculosis* H37Rv and *M. bovis* var BCG [77]	Temperate [77]	Yes [130]
Kerberos*/Siphoviridae* [131]	Capability to infect *M. tuberculosis;* presence of virion structure/assembly/nonstructural genes in its genome; having tRNA [131]	A/A2 [131]	Soil [131]	63.5 [131]	*M. smegmatis* mc^2^155 [131]	Temperate [131]	Yes [131]
Pomar16*/Siphoviridae* [131]	Capability to infect *M. tuberculosis;* virion structure/assembly/nonstructural genes in its genome; having tRNA [131]	A/A2 [131]	Soil [131]	63.5 [131]	*M. smegmatis* mc^2^155 [131]	Temperate [131]	Yes [131]
StarStuff*/Siphoviridae* [131]	Capability to infect *M. tuberculosis;* virion structure/assembly/nonstructural genes in its genome; having tRNA [131]	A/A2 [131]	Soil [131]	63.5 [131]	*M. smegmatis* mc^2^155 [131]	Temperate [131]	Yes [131]
Omega*/Siphoviridae* [132]	Lack of DNA ligase gene [133]; possible role in mycobacterial virulence as the phage encodes gene 61 that is a close homolog of tuberculosis Lsr2; may play role in humoral and cellular immune responses [134].	J [61]	Unknown	61.4 [61]	*Mycobacterium* sp. [132]	It is possibly temperate because it forms slightly turbid plaques, and stable lysogens could be recovered [135]	Yes [132]
Cjw1*/Siphoviridae* [136]	A possible role in mycobacterial virulence as the phage encodes gene 39, which is a close homolog of leprosy Lsr2; may play a role in humoral and cellular immune responses [137].	E [138]	Unknown	63.7 [61]	*Mycobacterium* sp. [136]	It is possibly temperate because it produces hazy to turbid plaques at 37 and 42 °C, respectively [139]	Yes [136]

## 3. Morphology

Mycobacteriophages are dsDNA viruses that could infect mycobacteria. Mycobacteriophages were primarily distinguished based on their morphologies via the electron microscope and their antigenic divergences [140,141]. In 1953, eleven mycobacteriophages were isolated and categorized based on their plaque morphologies and reactivity in serological and cross-resistance analyses [142]. However, according to the latest classification, mycobacteriophages were stratified into two prominent families of *Siphoviridae* and *Myoviridae,* including 11/210 and 5/87 subfamily/genera, respectively [143]. Members of these families are distinguished with morphological differences such as tail structure [143]. *Myoviruses* are members of cluster C phages and are characterized by their single genotypes, larger heads [144], and long contractile tails. *Siphoviruses* are genetically diverse and have long flexible non-contractile tails [8]. The application of transmission electron microscope (TEM) has facilitated the differentiation of mycobacteriophages in various families and clusters based on phages’ head and tail structures. The head in mycobacteriophage D29 (*Siphoviridae*) is almost isometric and it has a diameter of 65 nm, while the tail length is 150, 300, or 450 nm (Figure 1) [145]. The isometric head in TM4 has a diameter between 50 and 58 nm [146] that is linked to a flexible non-contractile tail with a 190 nm length [73] that ends in a bulb [146]. Tail length in mycobacteriophage Rahel (*Myoviruses*) is 89 nm, shorter and thicker than TM4 [147]. In both families of *Siphoviridae* and *Myoviridae*, tails are constituted of stacked rings of six subunits. In addition, genera differ by genome organization, DNA packaging mechanism, and presence or absence of DNA polymerase [148,149]. *Myoviruses* genera are susceptible to sudden temperature and osmotic changes [144]. The shape and size of capsids differ by the mycobacteriophage strains and their genome size, respectively. Most mycobacteriophages have an isometric capsid, with diameter ranges between 40 and 80 nm. Some mycobacteriophages, such as Corndog, Che9c, and Brujita [150], have prolate heads with length: width ratios between 2.5:1 and 4:1 [8]. Tail length and tip structure [150] impose another variation on the phage structure, in which the tail size ranges between 105 [43] and 350 nm [8].

## 4. Classification

The complete genome sequencing of mycobacteriophages such as L5 and D29 has made indispensable progress in bacteriophagology, paving the path for detecting and classifying other mycobacteriophages [83,98]. Mycobacteriophages are distributed into various clusters based on the level of similar nucleotides that existed in their genome sequences. Members of the same cluster have a genomic uniformity of above 35%, and a similarity of less than 35% would place a mycobacteriophage in another cluster [151]. Although mycobacteriophages of various clusters have some nucleotide sequences in common, their genetic matter underwent many small mutations such as insertion, deletions, and substitutions that are limited to a single gene or a small sequence of DNA [8]. Members of various clusters and even subclusters have different levels of homogeneity [150]. Genomic clustering via average nucleotide identities (ANI) revealed that the similarity among members of different subclusters ranged from 99.8% to 62.1%; however, the lowest homogeneity rate was seen among members of clusters J, K, O, Q, and V, with an average rate of 56.18% [150]. This is under the condition that a slight affinity was detected among clusters J, K, O, Q, V, and other clusters [150]. In dot plot analysis, mycobacteriophages with ANI values between 53% and 59% are not placed in the same clusters [150]. Additionally, Gepared analysis of tape measure protein (TMP) sequence in mycobacteriophages could give helpful information about cluster types of mycobacteriophages. TMP is a long and conserved gene that is presented in all mycobacteriophages. The result of Gepared analysis of 247 mycobacteriophages TMP sequences revealed that 98.8% of mycobacteriophages were comparably placed at a similar subcluster as the one predicted by whole-genome comparison analysis.

The size of the mycobacteriophage genome varies from 41 to 165 kb containing 50% to 70% guanine–cytosine (GC) [47]. To date, 17.44% of discovered mycobacteriophages have been sequenced, classifying them into 29 clusters (A–Z, AA, AB, and AC) [43,103] and 10 singletons (mycobacteriophages that are remotely related to the six-defined clusters) [47]. Twelve out of twenty-nine clusters (A–D, F–H, I, K–M, and P) were further divided into subclusters (a total of 71) based on the rate of average nucleotide identity (ANI) among the members of each category [8]. The majority of mycobacteriophages (*n* = 322) were characterized as cluster A, classifying them into 18 subclusters of A1-A14 and A16-A19 [8]. Cluster E is another large cluster (*n* = 52) with no subclusters recognized yet [8]. Some mycobacteriophages have no confirmative affinity to the other identified and sequenced mycobacteriophages. DS6A is one of the phages that has recently been classified as a singleton. Evidence shows that DS6A has an observable vicinity to both clusters of F and K since members of the two clusters produce similar integrase as DS6A [63]. However, mycobacteriophages classified in cluster F cannot replicate in *M. tuberculosis.* This indicates that members of cluster F and DS6A might be either derived from the same ancestor or coincidently induced by a mixed infection in another host bacterium (2016) [63]. In contrast, some mycobacteriophages have meaningful nucleotide comparability of above 90% with other identified mycobacteriophages, and because of that, they were categorized in the same clusters. For example, Donny belongs to subcluster B5 since its nucleotide sequences have the closest similarity to some members of subcluster B5, such as Acadin (99.99%) and Bae (93%) [52]. In addition, KingMidas has a genomic similarity of 99.1% with Scottish mycobacteriophage, a mycobacteriophage of cluster F, and because of that, it is fitted into cluster F [152]

## 5. Life Cycle of Mycobacteriophages

Analysis of the mechanism of gene expression in mycobacteriophages has been an upward trend in recent years. Two life cycles of lytic and lysogenic (temperate) were observed among members of various clusters. The life cycle of mycobacteriophages is influenced by the expression of specific genes at early and late gene transcription steps. The early gene transcription happens 30 min after insertion, whereas the late gene transcription begins 30 min after infection and takes around 180 min, leading to the lysis of the infected mycobacteria. Non-structural genes are expressed during early transcription, whereas the virion structure, integration of sequences of the genome, and lysis cassette are encoded at the late gene expression [8]. Some of the lytic mycobacteriophages that were commonly used in various studies are as follows: L5, D29, StarStuff, Kampy, and SWU1 in cluster A [8,40,41,42], Giles in cluster Q [153], Fruitloop in cluster F [154], and numbers of phages in cluster N [155]. The lysogenic life cycle was reported among mycobacteriophages belonging to clusters A, F, Q, and N [153,154,155,156].

## 6. Mycobacteriophages and Detection of Pathogenic Mycobacteria

The application of mycobacteriophages for diagnostic purposes has had an increasing trend in recent years. This is because most pathogenic mycobacteria are either slow-growing or have difficult growth requirements. Among these troublesome mycobacteria, *M. tuberculosis*, MAP, and *M. abscessus* are of more interest in diagnostic studies*. M. tuberculosis* is extremely slow-growing and typically involves the lungs and causes tuberculosis in humans [157] or occasionally affects the neural and skeletal systems as well as other organs [158]. MAP is another slow-growing mycobacterium and the etiologic agent of paratuberculosis, a gastroenteric condition that could involve ruminants (cattle and herd) [159,160]. MAP has a long generation time of 24 h [5], and this attribute elongates the speed of growth of the bacterium significantly. In contrast, *M. abscessus* is a fast-growing non-tuberculosis multi-drug resistant mycobacterium that is the causative agent of a broad range of infections, including skin, lung, soft tissue, and disseminated infections in humans [161]. The capability of mycobacteriophages in crossing the mycobacterial barriers has turned them into practical tools for diagnosing pathogenic and non-pathogenic mycobacteria. Some of the most-known phage-based diagnostic techniques are as follows: shuttle plasmids [7,19,72,162] and transduction of fluorescent or non-fluorescent foreign DNA into the mycobacterial genome and distinguishing the antibiotic resistance or viability of mycobacteria through fluorescent emission or formation of turbid lysogenic plaques [19,20,163]; phage amplification and detection of the viability of mycobacteria [24,25,164,165]; capture of viable target mycobacteria using mycobacteriophage proteins [166] or whole phages as ligands [79,87,88].

Mycobacteriophages are portable genetic structures that could be genetically manipulated and transferred to mycobacteria to either detect mycobacterial genes [72] or induce specific traits in the target bacteria and make them distinguishable from non-target bacteria. Mycolic acid is one of the major lipid components of the mycobacterial cell wall that consists of more than 60% of the cell wall structure [6]. The presence of this lipid in cell barriers could restrict the penetration and replication of exogenous DNA in the internal area of mycobacteria in normal conditions. This explains that mycobacterial cells should artificially become competent to adsorbing exogenous DNA and be genetically transformed. Liposome or polymeric-based transfection, viral vectors, electroporation, and shock waves are some techniques that make mycobacteria competent for transfection [167,168,169,170,171].

To date, the genomes of some mycobacteriophages have been used as vectors for cloning and making recombinant DNA using genetic engineering advancements [7]. In 1987, one of the first recombinant shuttle plasmids that could transduce larger fragments of foreign DNA into the target bacterial genome was engineered. This shuttle vector was a plasmid DNA of *Escherichia coli* that was fitted into the non-essential zone of TM4 mycobacteriophage [72]. This vector could be expressed not only in *E. coli* as a plasmid, but also as a phage in fast-growing mycobacteria such as *M. smegmatis*. However, this vector could not successfully be transfected in slow-growing mycobacteria such as *M. bovis* BCG strains and *M. tuberculosis* [72]. In 1988, mycobacteriophage L1 was manipulated to form shuttle plasmids containing cloned genes that could induce Kanamycin resistance in *E. coli* and *M. smegmatis*, in which turbid lysogenic plaques were produced [19]. In the recent study, to stably transfect the Kanamycin resistance gene into both fast- and slow-growing mycobacterial species, *M. smegmatis* and *M. bovis* BCG, respectively, transformed plasmids containing *M. fortuitum* (plasmid PAL5000) and *E. coli* (plasmid PIJ666) replicons and Kanamycin resistance gene were engineered and successfully injected to target mycobacteria via electroporation [19]. The production of shuttle plasmid, as an achievement, paved the path for the introduction of new diagnostic reporter genes [172,173] and transposons [174,175] or specific genes to different species of mycobacteria [162].

In 2019, mycobacteriophages were engineered containing fluorescent reporter genes of gfp, ZsYellow, and mCherry that could estimate the susceptibility of mycobacteria to antibiotics or drugs through the emission of fluorescent lights in the frame of fluorescence microscopy, flow cytometry, multi-well fluorimeter analysis. This innovation would sensitively facilitate the detection of multidrug-resistant mycobacteria, such as *M. tuberculosis*, which normally requires time, automated technologies, and abundant finance [163].

Furthermore, in a recent study in 2020, a recombinant TM4 mycobacteriophage containing a nanoluciferase (Nluc) reporter gene cassette was designed and transduced into different viable pathogenic, drug-sensitive, and drug-resistant auxotrophic strains of *M. tuberculosis* (Figure 2) [20]. The auxotrophic strains are usually used as model organisms for the assessment of drug-resistant *M. tuberculosis* and latent TB strains, in which these mutant strains underwent extensive in vivo or in vitro biosafety testing to be qualified for application in biosafety level 2 facilities [176]. Accordingly, drug-susceptibility tests (DST) were carried out, and the performance of transduced TM4-nluc in both viable drug-resistant and drug-sensitive *M. tuberculosis* strains was evaluated in the presence and absence of antibiotics, and a cellular limit of detection (LOD) of ≤ 10^2^ CFU was suggested for the assay (Figure 2) [20]. Interestingly, this analysis revealed that the expression of the nanoluciferase gene in drug-sensitive *M. tuberculosis* strains was restricted in the presence of some drugs corresponding with a reduction in generative light signals (Figure 2) [20]. However, some drug-resistant *M. tuberculosis* strains produced the maximum light signals even in the presence of antibiotics. Therefore, TM4-nluc would be a practical, quick, and economical tool for detecting antibiotic susceptibility in various strains of *M. tuberculosis* [20].

In an innovative study in 2019, mycobacteriophage D29 was exploited to lyse *M. smegmatis*, a mycobacterial model, and extract mycobacterial-induced topoisomerase IA (TOP1A) enzyme [177]. Later, the catalytic reactivity profile of TOP1A, a biomarker, was used to estimate the presence of mycobacterial species in crude samples. Furthermore, a specific mycobacterial TOP1A DNA substrate was designed to capture mycobacterial TOP1A in crude specimens via covalent bindings [177]. Afterward, transient TOP1A-induced 5′-phosphotyrosyl cleavages were fixed by catalytic ligation reactivity of TOP1A, and that was followed by the liberation of TOP1A and conversion of linear DNA substrate to circle. Frequent rolling circle amplification of circulated DNA substrate resulted in the generation of tandem repeats that could be tracked down to a single copy via fluorescent emission [177]. The assay detected a maximum of 6–9 × 10^5^ CFU/mL mycobacteria in samples proposing new perspectives for the detection of pathogenic mycobacteria such as *M. tuberculosis* [177].

Phage amplification-based techniques speed up the assessment of viability and antibacterial resistance in mycobacteria, specifically slow-growing species. Phage-infected mycobacteria were one of the first models used in antimicrobial resistance studies to disclose essential information about the therapeutic agents’ impact on slow-growing mycobacteria. Accordingly, in 1965 and almost for the first time, the impact of streptomycin on *M. smegmatis* infected with mycobacteriophage D28 was evaluated. The result revealed that the premature lysis of streptomycin-sensitive mycobacteria infected with D28 was promoted in the presence of streptomycin. In contrast, streptomycin did not have any impact on the replication of D28 in mycobacteria resistant to streptomycin [21]. In 1988, in a similar study, *M. aurum*—a fast-growing mycobacterium close to *M. tuberculosis* [22]—was infected with mycobacteriophage D29 and exposed to streptomycin along with other antituberculosis and antileprosy agents such as clofazimine, colistin, rifampicin, isoniazid, dapsone, and ethambutol, in which the result indicated that the 50% inhibitory concentration or minimal inhibitory concentration (MIC) ratio that could block the replication of mycobacteriophage D29 in *M. aurum* varied by the type of drug. This ratio was 1 μg/mL for streptomycin, clofazimine, colistin, and rifampicin. However, the ratio was higher for isoniazid, dapsone, and ethambutol, explaining that these drugs had no impact on the viability of *M. aurum* cells and the D29 life cycle [23]. The result of this study suggested that phage-infected bacteria would be practical models for evaluating the impact of therapeutic agents on slow-growing mycobacteria such as *M. ulcerans*, *M. paratuberculosis*, *M. leprae*, and *M. lepraemurium* [23].

Previous studies demonstrated that mycobacteriophage D29 (cluster A) could replicate in several viable fast- and slow-growing mycobacteria such as *M. smegmatis* [178]*,* MAP [87,179], and *M. tuberculosis* [180]. D29 was initially applied in the FASTPlaqueTB phage assay to detect the viability of *M. tuberculosis* complex in human sputum specimens taken from patients suspected of having tuberculosis in 2005 [164]. Between 2007 and 2021, D29 was exploited to determine the viability of MAP in dairy products [24], blood, and feces [25] within an adapted FASTPlaqueTB [24] assay, Actiphage core 2-day assay, or modified phage amplification techniques. From 2013 to 2020, phage amplification via D29 underwent modifications, including changes in the protocol of DNA extraction from lysed cells that were indicated with plaques (i.e., heating agar plaques [24]; purification of the extracted DNA from excised plaques via Zymoclean DNA Clean and Concentrator columns [181,182]); selective capture and concentrating MAP cells in samples via magnetic beads coated with MAP-specific complementary peptides (i.e., aMp3 and aMptD) and consequent magnetic separation. The recent modification led to the establishment of a technique that was named peptide-mediated magnetic separation (PMS) [165,183,184], constituting the following steps: (1) magnetic separation and capture of viable MAP cells, (2) D29 infection, (3) injection of phage DNA into the captured cells, (4) amplification of phage DNA in host cells, and (5) visualization step and quantification of MAP DNA via PCR [24] /qPCR *IS**900* (Figure 3) [165]. The mentioned steps enhanced the specificity and sensitivity of phage-based analyses in the detection of not only viable MAP but also other viable mycobacteria such as *M. bovis* in various sample types. In a study in 2016, phage amplification via D29 along with an isothermal DNA amplification by recombinase polymerase amplification (RPA) could effectively diagnose viable *M. bovis* BCG, a member of the *M. tuberculosis* complex group, in PBMC samples (isolated from bovine blood) within 48 h, in which the least concentration of the bacterium that the assay could detect in an artificially contaminated blood sample was 10 cells/mL and this point was estimated as the limit of detection for the assay (LOD) [185].

Visualization of phage-infected viable mycobacteria was another exciting improvement in phage amplification-based analysis. In 2017, viable MAP cells in milk samples were retrieved by peptide-mediated magnetic separation and infected with D29 accordingly. Then, to visualize the viability of MAP cells in samples, the lysate was cultured with a fast-growing mycobacterium, *M. smegmatis* on a solid medium, in which plaques that each corresponded to a viable MAP cell or clump of MAP cells formed around lysed MAP cells in a lawn of *M. smegmatis* (Figure 3) [165]. In the final step, to confirm the identity of the lysed plaques, DNA was extracted from 5–10 plaques and amplified by qPCR *IS*900 analysis [165]. This assay has been tested on various matrixes such as milk, feces, and blood [84,165,181]. In 2021, a study on the detection of viable circulating MAP in PBMC depicted that phage-qPCR could comparably detect a similar proportion of positive cases as fecal PCR and plasma antigen-specific IFN-γ among cattle involved with Johne’s disease (JD) [85].

On the other hand, mycobacteriophages and their expressed proteins can efficiently capture viable mycobacteria such as MAP, *M. tuberculosis*, and *M. smegmatis* in various specimens. In 2014, an analysis of mycobacteriophage L5 revealed that the phage genome encodes proteins that could be used as potential ligands to capture viable MAP and *M. smegmatis* [166]. For instance, immobilized tail protein (Gp6) could capture both MAP and *M. smegmatis* in samples, and a lysine protein (Gp10) binds more specifically to *M. smegmatis* [166]. Experiments on Gp6 and Gp10 demonstrated that these proteins could specifically target MAP and *M. smegmatis,* neither other mycobacterial species such as *M. marinum* nor Gram-negative bacteria such as *E. coli*, salmonella, and campylobacter. However, Gp6 could also bind to some chemically-synthetized superficial mycobacterial glycans, which may undermine its specificity [166]. In 2020, in an innovative study, mycobacteriophage D29 was directly coupled to magnetic beads via covalent bonds to capture MAP in milk samples through a magnetic separation step [87] (Figure 4). This assay was called phagomagnetic separation-qPCR assay [87]. On another level, recovered viable MAP cells were resuspended in Middlebrook (MB) 7H9 supplemented with 10% Oleic Albumin Dextrose Catalase (OADC) and 2 mM CaCl_2_, and that was followed by the incubation of suspensions at 37 °C/2 h and a thermal shock at 55 °C/1–2 min [87,88]. Later, the presence of MAP DNA in lysates (Figure 4) was assessed by qPCR *IS**900* analysis [87]. The modified phage assay introduced a meaningful sensitivity along with speed to the procedure of MAP viability assessment, in which the length of diagnosis decreased from 48 h (PMS-phage assay) to almost 7 h with a LOD of 10 MAP cells per 50 mL [87] or 10 mL [88] milk. Although any viable mycobacterial species that could be infected with D29 might be recovered, and their DNA would be transferred to the lysate, this is the qPCR analysis that influences the specificity of the assay remarkably. Accordingly, in the assessment of the efficiency of three qPCR methods, including SYBR Green (SensiFAST™ SYBR^®^ Hi-ROX Kit, Bioline Reagents Limited, London, UK) qPCR *IS*900, TaqMan (SensiFAST™ Probe^®^ Hi-ROX Kit, Bioline Reagents Limited, London, UK) qPCR *IS*900, and Techne™ PrimePro qPCR DNA detection kit (Techne™, Staffordshire, UK) for detection of MAP DNA presented in the same lysate, the diagnostic level considerably improved through TaqMan qPCR *IS*900 analysis rather than other qPCR methods [87]. In 2020 and 2021, phagomagnetic separation-qPCR assay was tested on bovine [87] and sheep/goat [88] milk samples, and it sensitively detected viable MAP in 49% out of 100 and 48.78% out of 41 of the studied animals, respectively [87,88].

In 2021, mycobacteriophage SWU1 was similarly used as ligands to coat magnetic beads and retrieve viable *M. smegmatis* in samples. *M. smegmatis* was selected as a model *Mycobacterium* sp. in this analysis since it is fast-growing and resembles the pathogenic mycobacteria (i.e., *M. tuberculosis*) in terms of physiological characteristics [79]. Therefore, the numbers of viable *M. smegmatis* cells in samples were estimated through the quantification of bioluminescent signals emitted from intracellular adenosine triphosphate (ATP) during lysis (after 60 min replication) of viable cells that were already captured and infected with SWU1 [79]. This method effectively detected viable *M. smegmatis* in various human specimens such as saliva, urine, and serum at a minimum concentration of 3.8 × 10^2^ CFU mL^−1^ (this point was adjusted as the limit of detection (LOD) of the assay) [79].

## 7. Mycobacteriophages and Treatment of Mycobacterial Infections

Mycobacterial-associated infections have remained a serious concern for decades [186]. Executing screening measures and appropriate antibiotic therapies plays a critical role in treating these infections [3]. However, the emergence of multi-drug resistant mycobacterial species undermined the effectiveness of the current antibiotics/drugs remarkably [187].

Nowadays, infections caused by *M. tuberculosis*, *M. avium*, and *M. abscessus* [188] exposed humans, specifically immunodeficient patients (i.e., AIDS), to life-threatening conditions. For example, in 1998, the administration of protease inhibitors for treating infections caused by human immunodeficiency virus type 1 (HIV-1) inhibited *M. avium* bacteremia in these patients [189]. However, anti-HIV-1 drugs lost their efficacy against MAP infection in these patients soon after developing resistance to the drugs [190]. Therefore, phage therapy has opened new horizons in treating mycobacterial infections insensitive to antibiotic treatment. Intact/genetically modified mycobacteriophages and their products, such as lysin, in isolation or within a synergistic antibiotic therapy, influence the life cycle of mycobacteria through either lysing the host cells or inducing mutations to mycobacterial genomes, and that would be followed by disruption of mycobacterial replication in host cells. Among the mycobacteriophages that have been isolated from different environmental or clinical sources, thirteen phages (DS-6A, TM4, D29, T7, P4, PDRPv, BTCU-I, Bo4, SWUI, GR-2I/T, My-327, Ms6, and Bxz2) have therapeutic potentials against *M. tuberculosis* and some other mycobacteria [191]. In 2018, a survey was conducted by the center of Innovative Phage Applications and Therapeutics (IPATH) on existing bacteriophages appropriate for phage therapy, in which 90 mycobacteriophages were tested against various mycobacterial species as follows: 47 against *M. abscessus*, 23 against *M. avium*, 7 against *M. chimera*, 7 *M*. species, 2 *M. chelonae*, 1 *M. smegmatis*, 1 *M. xenopi*, 1 *M. bolletii*, and 1 *M genavense* [192]. Among the bacteriophages selected for treatment of mycobacterial infections, nine and four lytic phages functioned against *M. abscessus* and *M. chimera,* respectively, in which they were administrated to four and one patients suffering from infections caused by *M. abscessus* and *M. chimera,* respectively [192]. Overall, 17 out of 119 patients underwent intravenous phage therapy with selected bacteriophages, and the treatment was effective on 7 out of 10 patients who experienced severe infections resistant to antibiotics [192].

In 1981, in a primary study on the impact of phage therapy on healing tuberculosis, three different phages, DS-6A, GR-21/T, and My-327, were subcutaneously administrated (10(6)/1 mL) to guinea pigs that were artificially infected with *M. tuberculosis* strain H 37 Rv for 10 days [65]. The result showed that all three phages deteriorated the number of bacilli in the studied animals, which corresponded with positive changes in spleen and hilus indices [65]. However, DS-6A seemed to function more effectively than the two other phages since its therapeutic impact on the spleen index was more meaningful (0.19) [65]. The recent reduction in the number of *M. tuberculosis* bacilli after phage therapy could be explained by the fact that the free bacilli can circulate in the bloodstream and be infected by phages [193], in which the immune cells such as macrophages can recognize these phage-infected *M. tuberculosis* bacilli and phagocytize them consequently. By liberating the mycobacteriophages trapped in macrophages, the intracellular bacilli would get involved in infection [193]. This hypothesized mechanism could explain how phage-associated infection disseminates in model animals and reduce the number of bacilli in the bloodstream [193].

On the other hand, mycobacteriophages could impede or destroy the vital enzymatic pathways in the host mycobacteria. Citrate lyase is an essential enzyme in eukaryotes, prokaryotes, and archaea that transforms citrate to acetyl-CoA [194], a cofactor that plays an important role in the oxidative pathway of fatty acids, carbohydrates, amino acids, and Krebs cycle [195]. In bacteria, this enzyme has three subunits, including CitD, CitF, and CitE. However, *M. tuberculosis* does not have CitD and CitF subunits in its genome, and instead, it expresses two homologous subunits for CitE (CitE1 and CitE2) [194]. These two subunits could guarantee the normal replication of *M. tuberculosis bacilli* internalized in macrophages. However, research in 2018 showed that temperature-sensitive mycobacteriophages could impose mutations on CitE subunits (one or both) and disrupt CitE functionality [194], in which *M. tuberculosis* strains that underwent mutations in these two subunits would be sensitive against oxidative stress. In animal models (guinea pig), this double mutation also declined the bacterium’s growth in the lungs and spleen [194]. The recent functionalities from temperature-dependent mycobacteriophages in suppressing the replication of *M. tuberculosis* in macrophages could be utilized to treat infections caused by the bacterium [194].

Today, mycobacteriophages such as ZoeJ, BPs, and Muddy, individually or within phage cocktails along with antibiotic therapy, have shed new hopes on the possibility of treating multidrug-resistant infections caused by *M. abscessus, M. tuberculosis*, and *M. avium*. ZoeJ (a close relative of mycobacteriophage TM4) is one of the most infective mycobacteriophages against a wide range of mycobacteria such as *M. tuberculosis*, *M. smegmatis*, and *M. avium;* therefore, it would be a potential option in phage therapy [59].

In 2021, the result of a clinical trial on a patient with cystic fibrosis (CF) suffering from a prolonged disseminated infection with *M. abscessus*, *Pseudomonas aeruginosa,* EBV viremia (8 years), with a history of two lung transplantations along with administration of various oral/intravenous antibiotics, lumacaftor/ivacaftor, and immunosuppressive drugs (e.g., Clofazimine, bedaquiline, Mycophenolate mofetil, iv rituximab) revealed that the hypodermic administration of mycobacteriophages cocktail containing Muddy (wild type) and engineered BPs33ΔHTH-HRM10 and ZoeJΔ45 every 12 h for seven months (10^9^ pfu/dose of each phage) could improve the recovery in this patient [16,17,196]. Furthermore, regarding the engineered phages used in the recent study, repressors that induce the lysogenic life cycle in these phages were deleted, and these phages were converted to lytic phages. Accordingly, this analysis indicated that the application of mycobacteriophage cocktails could resolve problems caused by phage-resistant *M. abscessus* strains and temperate phages that were previously integrated into the bacterial genome and contributed to expanding the infection [17].

Evidence reveals that most *M. abscessus*-associated infections are multidrug resistant [16]. *M. abscessus* strain GD01 is one of the most-used strains successfully infected with and eliminated by mycobacteriophages such as Muddy, BPs, and ZoeJ [196]. However, *M. abscessus* subsp. *bolletti* F1660 could only be responsive to one mycobacteriophage [17]. In 2021, in vitro analysis of GD01 infected with Muddy depicted that the administration of standard drugs could enhance the efficiency of cell lysis via Muddy. This functionality was also observed in an in vivo survey on CFTR-depleted zebrafish embryos, which received both Muddy microinjections and antibiotic treatment, leading to a simultaneous unprecedented surge and decline in the viability of larvae and pathological manifestations, respectively [16]. However, Muddy was unable to react against macrophage-ablated larvae since macrophages are the primary cell line that, via the formation of granuloma, play an essential role in the control of *M. abscessus* infection, in which loss of macrophages accelerated the death of larvae several days after infection. This underlines the importance of functional innate immunity in a thriving phage therapy [16]. Surprisingly, synergistic therapies via mycobacteriophages and antibiotics could improve the quality of treatment in confronting fast-growing mycobacteria such as *M. abscessus*. This would be associated with a remarkable reduction in the length of therapy, portion of antibiotics consumed in each interval, and side effects of antibiotics [17]. *M. tuberculosis*-associated infections would also be responsive to the synergistic therapies based on the simultaneous administration of both mycobacteriophages and antibiotics. In 2016, an experimental analysis of *M. smegmatis,* model mycobacterium, that underwent a synergistic treatment with SWU1gp39 and antibiotics, including isoniazid, erythromycin, norfloxacin, ampicillin, ciprofloxacin, ofloxacin, rifampicin, and vancomycin, revealed that the permeability of *M. smegmatis* to the mentioned antibiotics increased after infecting mycobacteria with SWU1gp39 [18]. Additionally, two studies in 2019 demonstrated that mycobacteriophages in whole and their induced lysins could be exploited to treat mycobacterial infections. Some bacteriophages produce lysing enzymes that could collapse the extracellular matrix in biofilms and coney antibiotics to bacteria [197]. Interestingly, lysin B-derived mycobacteriophage D29 could avert the dissemination of infection caused by *M. ulcerans* in the footpad of a mouse model [56].

Several studies have been conducted to unveil whether or not changes imposed on mycobacteriophages through synergistic therapies could affect the phage infectivity or even the efficiency of phage therapy. Accordingly, a study on the assessment of various pathophysiological conditions such as pH, low growth rate, and hypoxia on the dynamic of infections induced by a phage cocktail in mycobacteria such as *M. smegmatis* depicted that the phage cocktail could sustain various conditions for a long period without any changes that affect their activities in host cells [15]. Further investigation demonstrated that the studied mycobacteriophages could efficiently replicate in *M. tuberculosis,* regardless of whether bacilli are in lag or logarithmic growth phase [15]. Mycobacteriophages with antibiotics such as isoniazid and rifampicin could be a treatment choice for infections caused by drug-resistant mycobacterial species [15]. This is under the condition that the infectivity of engineered mycobacteriophage phAE159 and wild-type phage D29 against *M. tuberculosis* could be attenuated when they were simultaneously administrated with aminoglycoside antibiotics such as kanamycin, hygromycin, or streptomycin [198]. However, the recent phages were still infective in combination with spectinomycin [198]. This result suggested that the amino sugar group in these antibiotics could restrict the replication of mycobacteriophages DNA in *M. tuberculosis* [198].

To the best of our knowledge, the main challenges that phage therapy faces are either the insensitivity of the dormant mycobacteria (i.e., *M. tuberculosis* and *M. avium*) to common treatments or the inaccessibility of phages to intracellular mycobacteria that escaped from the immune system and concealed in macrophages.

Application of non-pathogenic mycobacteria infected with lytic mycobacteriophages against pathogenic mycobacteria (i.e., *M. tuberculosis* and *M. avium*) engulfed in macrophages would be an innovative solution for deceiving immune cells and transferring mycobacteriophages into phagocytes [193]. Furthermore, this method could remarkably enhance the exposure of intercellular mycobacteria to bacteriophages and consequent lysis of them since phages could not easily pass the cell barriers as well as antimicrobial agents [193]. The original study reported that TM4-infected *M. smegmatis* could successfully diminish the number of viable *M. avium* or *M. tuberculosis* in infected macrophages (RAW: peritoneal cell line in mouse) within a specific time if sufficient loads of the mycobacteriophage were exploited [193].

The other solution to face intracellular mycobacteria that are inaccessible-to-mycobacteriophages is encapsulating phages in liposomes. Under this circumstance, the entrance of phages to the internal area of immune cells in hosts would be facilitated, and mycobacteriophages could be concealed from neutralizing antibodies and acidic pH in the stomach [17,199,200]. Moreover, encapsulated phages could be applied in various forms such as an inhaler, oral, subcutaneous or topical, intramuscular, and intravenous administrations [17]. Previously, mycobacteriophages such as TM4 were capsulated in a large Amikacin liposome inhaler and administrated to patients suffering from NTM-lung infection to enhance the sputum conversion rate [201]. Inhalation of encapsulated mycobacteriophages could further influence the efficiency of phage therapy, in which a study on the impact of three different inhalation systems (vibrating mesh nebulizer, jet nebulizer, and soft mist inhaler) on the output phage concentration and rate of delivery of encapsulated aerosolized saline mycobacteriophage D29 revealed that no significant reduction in the titers of D29 happened after aerosolization by vibrating mesh nebulizer and soft mist inhaler devices (*p* > 0.1). In contrast, aerosolization of D29 via jet nebulizer reduced D29 concentration remarkably (*p* < 0.0005) [202]. In addition, the speed of D29 delivery via vibrating mesh nebulizer was 6000 times higher than jet nebulizer, which privileged mesh inhaler nebulizer over other inhaler systems [202].

Besides the advantages of phage therapies in improving the quality and speed of treatment of mycobacterial infections, some measurements should be taken into account in confronting unpredicted challenges. The significant difficulties in phage therapy appear when it is applied on scales larger than clinical trials. Under this circumstance, phage therapy relies on isolating more phages, developing screening techniques that quickly distinguish the therapeutic phages from others, developing efficient phage-based treatment strategies that can be effective against biofilms, establishing a safe and certified phage preparation step for production and formulation of phages in larger scales, providing the condition that guarantees the stability of phages during the storage and transportation [203].

## 8. Phage Resistance

Phage resistance is a condition in bacteria in which a phage cannot induce the expected infection response in its bacterial host anymore. Bacteria follow various mechanisms to resist phage infection, including modifications or concealments of phage receptors as a result of mutations, damages to phage DNA by nucleases or restriction enzymes, and the CRISPR-Cas system [204]. In Gram-positive bacteria, phage resistance is associated with any alterations in the structure of polysaccharides, teichoic acids, and outer membrane proteins that impede phage adsorption. Phage-resistant bacteria become less virulent or more susceptible to antibiotics in charge of having an equivalent metabolism that favors various environmental conditions [205]. Further analysis of the susceptibility of *M. abscessus* isolates to phage treatment revealed that the colony morphology (rough or smooth) significantly influences the resistance of *M. abscessus* to phage treatment, in which strains that developed rough colonies were more likely (80%) to be infected with and eliminated by at least one mycobacteriophage [206]. However, smooth-colony strains were responsive to none of the selected mycobacteriophages [206]. In fact, the formation of rough colonies in some strains of *M. abscessus* is induced by insertions and deletions in *mps1* and *mps2* genes that encode the synthesis of glycopeptidolipids. However, mutations in polyketide synthesis, *uvrD2*, and *rpoZ* genes could cause resistance in strains that produce smooth colonies [206]. Moreover, *M. abscessus* strains are prone to the genetic discrepancy directed by prophage and plasmid mobilome. Since prophages and plasmids have a high profusion and variation and carry a broad repertoire of genes, these elements could affect the susceptibility, virulence, and defense of *M. abscessus* strains to mycobacteriophages [207]. In 2021, a study on 82 clinical isolates of *M. abscessus* demonstrated that the majority of the strains (85%) had one or more prophages containing sequences from a minimum of 17 clusters that were attached to 18 various *attB* sites. These prophages induced the production of 19 sets of polymorphic toxin and toxin-immunity systems, each marked by WXG-100 proteins and transported through secretion system type VII [207].

Interestingly, *M. smegmatis* and *M. tuberculosis* employ five strategies to escape bacteriophages and avoid infection, including unpredictable target specificity and counter-defense measures, single-subunit restriction system, heterotypic exclusion system, predicted (p) ppGpp synthetase, and prophage-mediated viral defense [155], however, phages could also detour these tactics [2]. Being a temperate phage could also impede the entry of invasive phages to host cells. Studies demonstrated that genomes in mycobacteria, specifically fast-growing ones, contain many prophages. However, prophages in mycobacterial species isolated from clinical specimens carried more virulent genes than environmental species [94].

## 9. Conclusions

The exploitation of mycobacteriophages in medicine opened new horizons in diagnosing and treating mycobacterial-associated infections, specifically those caused by *M. tuberculosis*, *M. avium* spp., and *M. abscessus*. The properties that make mycobacteriophages potential candidates for phage-based diagnosis and therapy against mycobacteria are related to their capability in selective transduction of foreign DNA into the mycobacterial genome, formation of superinfection stable lysogens and induction of toxicity against both slow- and fast-growing mycobacteria, production of endolysins (Lysin A, Lysin B, or both) destructive to the mycobacterial cell wall, transportation of small mobile genetic elements, reduction of the acid-fastness of specific members of mycobacteria, excision of the antimicrobial resistance genes, lysis of non-pathogenic or pathogenic mycobacteria in various environments such as the bloodstream and liposomal macrophages. Previous studies depicted that mycobacteriophages in the frame of recombinant shuttle plasmids containing fragments of cloned foreign DNA, antibiotic resistance genes, or nanoluciferase (Nluc) reporter gene cassette could discriminate between drug-sensitive and drug-resistant pathogenic mycobacterial species. In addition, phage amplification-based techniques disclosed essential information about the impact of therapeutic agents on slow-growing mycobacteria (i.e., MAP, *M. tuberculosis*) and their viability in various sample types such as milk, blood, feces, tissue, etc. Importantly, mycobacteriophages and their expressed proteins could be potential ligands for the capture and detection of viable mycobacteria. On the other hand, intact/genetically modified mycobacteriophages and their products, such as lysin, in isolation or within a synergistic antibiotic therapy, demonstrated therapeutic impacts on infections caused by *M. tuberculosis* and *M. abscessus*. Mutations induced by mycobacteriophages could also have therapeutic applications, in which mutations in subunits of citrate lyase in *M. tuberculosis* strains make the bacterium susceptible to oxidative stress and disrupt the normal replication inside macrophages. However, mycobacteria always have different direct or indirect mechanisms to escape infection via mycobacteriophages. One of these strategies that impose struggles on phage therapy is the presence of mycobacteria engulfed in macrophages. Application of non-pathogenic mycobacteria that have already been infected with lytic mycobacteriophages or mycobacteriophages encapsulated in liposomes would be promising solutions to this problem, in which they enable mycobacteriophages to cross barriers that exist on the surface of the phagocytic cells and get access to the internalized mycobacteria. Nevertheless, strategies that immunize mycobacteria against infections might fail by concurrent administration of phage cocktails and antibiotics to patients. On the other hand, synergistic therapy might also have unpredicted consequences since patients must take multiple drugs in various quantities and titers of mycobacteriophages. Therefore, careful studies are needed to elucidate the side effects of these combinatory treatments. This should be along with discovering new mycobacteriophages specific to the most struggling pathogenic mycobacteria and finding alternative therapies that have the maximum efficacy and least side effects for patients.

## Figures and Tables

**Figure 1 pathogens-11-00777-f001:**
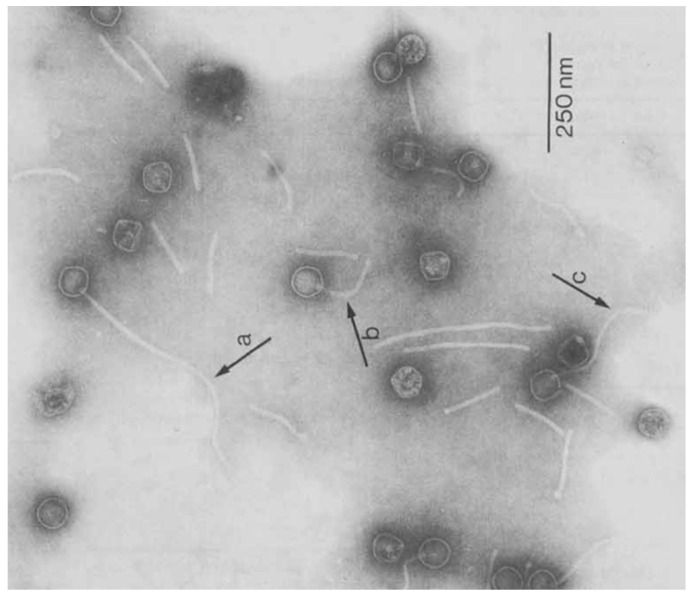
TEM figures of mycobacteriophage D29; a, b and c arrows demonstrate the variation of tail length in mycobacteriophage D29 [145].

**Figure 2 pathogens-11-00777-f002:**
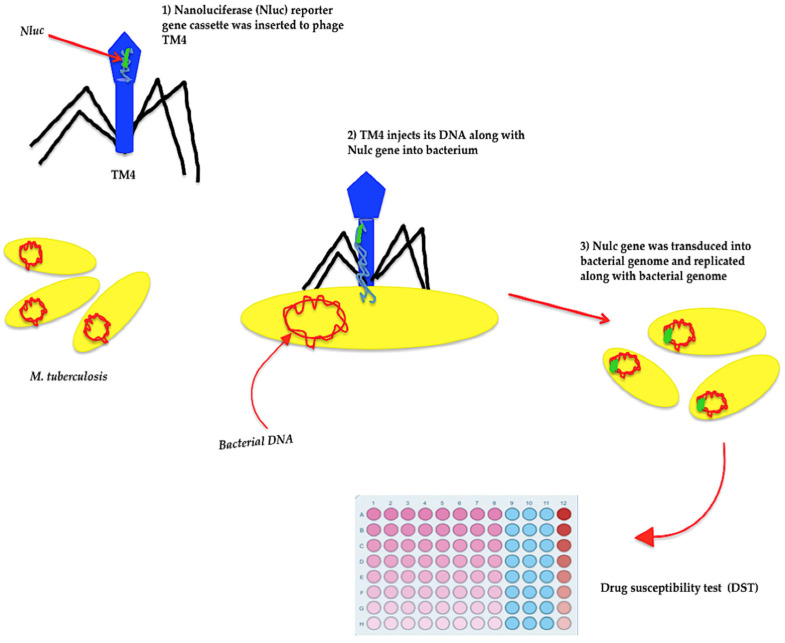
Detection of viable drug-sensitive/resistant *M. tuberculosis* via transducing TM4 containing a nanoluciferase (Nluc) reporter gene cassette into bacterial genome. Note: The sizes are not realistic in this figure, and all components were magnified to make the procedure more understandable.

**Figure 3 pathogens-11-00777-f003:**
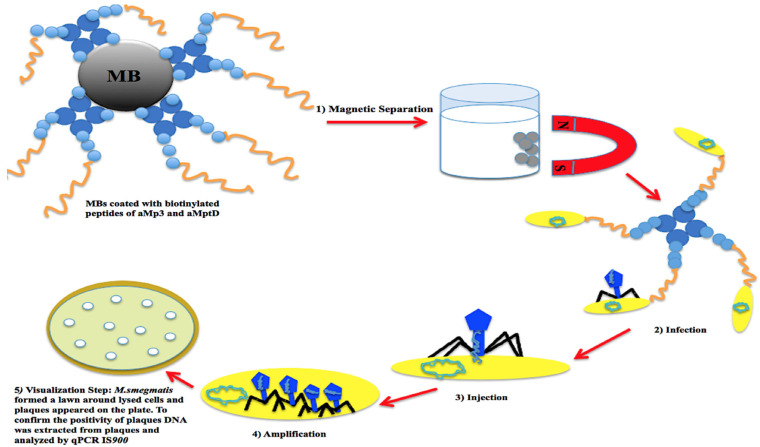
The procedure of peptide-mediated magnetic separation (PMS) phage assay and detection of the viability of MAP in milk samples (MB: magnetic bead). Note: The sizes are not realistic in this figure, and all components were magnified to make the procedure more understandable.

**Figure 4 pathogens-11-00777-f004:**
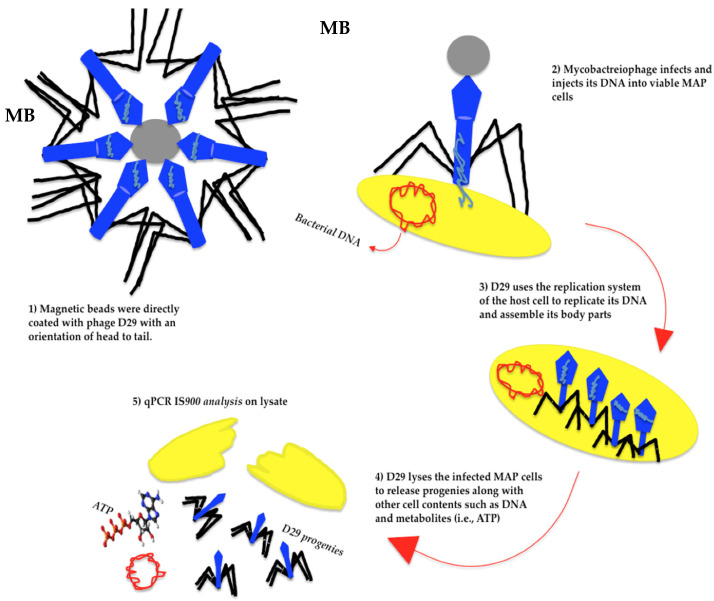
The procedure of evaluating MAP viability via phagomagnetic separation-qPCR (MB: magnetic bead). Note: The sizes are not realistic in this figure, and all components were magnified to make the procedure more understandable.

## Data Availability

Not applicable.

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
