# Peer review of "A Review on Mycobacteriophages: From Classification to Applications"

_pathogens, 2022, doi:10.3390/pathogens11070777_

Round 1

Reviewer 1 Report

This is a manuscript titled “An update on Mycobacteriophages”. The word “update” implies something new or recent, but most of the information contained in the paper are relatively old and already published somewhere else.

The draft is incredibly long, and it contains some very detailed explanations that I don’t think are necessary. Paragraph two (“a brief history about ….”) is not brief at all and it can be easily deleted as it doesn’t add much to the update. This looks more like a book chapter rather than an article.

The conclusion can also be significantly simplified as it should simply summarize the main concepts in a logical way, rather than another long list of different phages.

There is also a significant focus on M. smegmatis, both in the text and in the table, but this is a non-pathogenic mycobacterium and such incredible amount of data is not that relevant from a clinical point of view.

Table 1 should be completely changed, to list the phages active against clinically relevant mycobacteria (M. tuberculosis, M. avium and M. abscessus) and not an extensive list of phages against M. smegmatis.

In addition, there are numerous grammatical and structural mistakes when writing in English and the manuscript would certainly benefit from an expert translation service.

Author Response

We thank #Reviewer 1 for the helpful suggestions and comments, here our response point by point:

Comment 1: This is a manuscript titled “An update on Mycobacteriophages”. The word “update” implies something new or recent, but most of the information contained in the paper are relatively old and already published somewhere else.

Answer: To eliminate ambiguity, we replaced the word “update”” with “review” in the title “A Review about Mycobacteriophages; From Classification to Applications”.

Comment 2: The draft is incredibly long, and it contains some very detailed explanations that I don’t think are necessary. Paragraph two (“a brief history about ….”) is not brief at all and it can be easily deleted as it doesn’t add much to the update. This looks more like a book chapter rather than an article.

Answer: This part of the manuscript contains some practical information about mycobacteriophages infective to M. smegmatis, M. tuberculosis, M. avium subsp. paratuberculosis, and M. abscessus presenting their diagnostic and therapeutic properties. Extra explanations were removed to consider your viewpoint and not to deviate from the aim of this section.

Comment 3: The conclusion can also be significantly simplified as it should simply summarize the main concepts in a logical way, rather than another long list of different phages.

Answer: The conclusion has been summarized as you recommended. 

Comment 4: There is also a significant focus on M. smegmatis, both in the text and in the table, but this is a non-pathogenic mycobacterium, and such incredible amount of data is not that relevant from a clinical point of view.

Answer: M. smegmatis is a non-pathogenic mycobacterium that could easily get infected with a wide range of mycobacteriophages. This bacterium has primarily been used as a standard and safe mycobacterium species in almost all phage-based studies. Pathogenic mycobacteria were always the second-line candidates in these studies. In addition, much of our knowledge related to the diagnostic and therapeutic aspects of these mycobacteriophages was acquired because M. smegmatis was the host bacterium. These results could be expanded to other mycobacteria in some cases. We believe that introducing mycobacteriophages active to M. smegmatis and talking more about their therapeutic/ diagnostic capabilities would motivate researchers to either evaluate the infectivity of these phages against pathogenic mycobacteria or manipulate/ transform these phages in the way that they could also infect pathogenic mycobacteria.

Comment 5: Table 1 should be completely changed, to list the phages active against clinically relevant mycobacteria (M. tuberculosisM. avium and M. abscessus) and not an extensive list of phages against M. smegmatis.

Answer: The table 1 was modified as you recommended.

Comment 6: In addition, there are numerous grammatical and structural mistakes when writing in English and the manuscript would certainly benefit from an expert translation service.

Answer: The manuscript underwent extensive changes, and it was polished grammatically.

Again thank you for your help, we do believe that the manuscript now is suitable for publication.

Yours

Leonard A Sechi

Reviewer 2 Report

Thank you for preparing this comprehensive review on mycobacteriophages. It has a different and broader scope than other recent reviews on phage-based diagnostics for MAP, for example.  It gets a bit tedious in parts (classification section for instance), but overall I think it will be of value to mycobacteriologists once revised and improved.  I think the content is largely fine, but overall the English needs considerable attention in terms of grammar, sentence construction, paragraphing, etc. and you should seek the help of a native English speaker to resolve this major deficiency of the manuscript. 

I think Conclusions section needs to be shortened, and I don't think two figures are necessary - Fig 5 and 6. 

I have annotated the manuscript file with comments and corrections for your attention.  The manuscript is not suitable for publication in its current form. 

Author Response

We thank #Reviewer 2 for his helpful comments and suggestions. We do believe that the manuscript is now improved and suitable for publication under your guidance.

Comment 1: Thank you for preparing this comprehensive review on mycobacteriophages. It has a different and broader scope than other recent reviews on phage-based diagnostics for MAP, for example.  It gets a bit tedious in parts (classification section for instance), but overall I think it will be of value to mycobacteriologists once revised and improved.  I think the content is largely fine, but overall the English needs considerable attention in terms of grammar, sentence construction, paragraphing, etc. and you should seek the help of a native English speaker to resolve this major deficiency of the manuscript. 

Answer: Thank you very much for your helpful comments. The manuscript underwent extensive revision, and your suggestions were taken into consideration. The manuscript was also polished grammatically.  

Comment 2: I think Conclusions section needs to be shortened, and I don't think two figures are necessary - Fig 5 and 6. 

Answer: The conclusion was summarized as you recommended and figure 5 and 6 were as removed accordingly.

Comment 3: I have annotated the manuscript file with comments and corrections for your attention.  The manuscript is not suitable for publication in its current form. 

Answer: All your concerns were carefully assessed and considered to improve the manuscript. We hope that you find the modifications interesting.

Annotations:

  1. Line 10: This is first mention so should be in full here and abbreviated as M. at subsequent mentions.

Answer: Done

  1. Line 10: add s to million and add word of

Answer: Done

  1. Line 15: spelling - capital M for Mycobacterium and add an o to tuberculosis

Answer: Done

  1. Line 16: English needs attention here

Answer: the sentence changed.

  1. Line 22: Reword this text... such as DS6A that has been shown to be able to infect members of the M. tuberculosis complex.

Answer: Done

  1. Line 24: add comma here and delete comma after word those.

Answer: Done

  1. Line 31: add an s to include

Answer: Done

  1. Line 35: which study or studies are you referring to here? citation missing

Answer: Citation was added. Line 38-39: while others have been isolated using other mycobacterial species such as vB_MapS_FF47 [5] that can infect MAP ATCC19698 [5].

  1. Line 38: add s to human

Answer: Done

  1. Line 39: Strikethrough “Kind”

Answer: Done

  1. Line 42: Strikethrough “of”

Answer: Done

12: Line 43: suffering not suffered

Answer: Done

13: Line 49: should this read modify or degrade drugs?

Answer: Yes, it has been corrected. (Line 50)

  1. Line 65: should be subspecies not spp. (which means species)

Answer: Done

15: Line 87: you probably mean donor

Answer: it was corrected. 

  1. Line 97: is spelling correct? should it be y instead of first i?

Answer: This part was deleted. Some modifications were implemented in “brief history” section to summarize it. To find the track of changes, please refer to the revised version of the manuscript.

  1. Line 102: add an s

Answer: Done

  1. Line 131: presumably this should be Lysin B

Answer: It was corrected.

  1. Line 137: arise not raise

Answer: It was corrected.

  1. Line 156-157: You can substitute M. for all Mycobacterium in this sentence

Answer: Done

  1. Line 162: Strikethrough “ycobacterium”

Answer: Done

  1. Line 164: which agar are you referring to here?

Answer: In fact, it was Middlebrook broth that was supplemented with some reagents. it was further described as follows: “Myco agar including 4.7 g Middlebrook 7H9, 5 g nutrient broth, 10 mL 50% glycerol, 0.05% Tween 80 per liter, and 1.2% agar  supplemented with 20 μg/mL Kanamycin [4])”

  1. Line 168: usually lower case m and c. Please check every instance of this strain designation is corrected throughout the manuscript

Answer: They were corrected throughout the manuscript.

  1. Line 168: Strikethrough “The all” and

Answer: Done, “The” was removed.

25: Line 177: studies rather than researches (which isn't an English word)

Answer: It was replaced with studies.

  1. Line 187: where are these clusters you are referring to defined? Can you cite a database or other source - maybe The Actinobacteriophages Database at PhagesDB.org

Answer: The following citation was added: Delesalle, V.A.; Tanke, N.T.; Vill, A.C.; Krukonis, G.P. Testing Hypotheses for the Presence of TRNA Genes in Mycobacteriophage Genomes. Bacteriophage 2016, 6, e1219441–e1219441, doi:10.1080/21597081.2016.1219441.

  1. Line 197: spelling incorrect

Answer: This part was removed

  1. Line 215: Nassau I think

Answer: It was corrected

  1. Line 227: Has this acronym been defined earlier in text, i.e. after first mention of Mycobacterium avium subsp. paratuberculosis?

Answer: Yes. It is defined in the abstract

  1. Line 242: belonging

Answer: It was corrected.

  1. Line 249: should this be tuberculous

Answer: It was corrected as follows: M. tuberculosis complex (MTBC) such as M. tuberculosis and non-tuberculosis mycobacteria (NTMB) such as M. avium and M. fortuitum [67].

  1. Line 277: what is meant by exactly?

Answer: It was changed to specifically

  1. Line 324: too late for acronym here. It needs to come earlier in text

Answer: The name of mycobacterium was removed from the line and acronym was defined in the abstract.

  1. Line 324: you can't start sentence with word as

Answer: It was corrected throughout the manuscript. The related sentence was removed in the revised version.

  1. Line 328: these rather than this?

Answer: In fact, D29 was meant. It was corrected.

  1. Line 329: commas between sample types rather than /

Answer: It was corrected and slash was replaced with commas

  1. Line 386: this sentence doesn't constitute a new paragraph

Answer: It was fitted into the previous paragraph.

  1. Line 441:no new paragraph here.

Answer: It was corrected

  1. Line 456: has been

Answer: It was corrected

  1. Line 469: Strikethrough “is”

Answer: Done

41: Line 461: Strikethrough “2009”

Answer: Done

  1. Line 461: Strikethrough “2003”

Answer: Done

  1. Line 463: Strikethrough “2020-2021”

Answer: Done

  1. Line 464: Strikethrough “2014”

Answer: Done

45: Line 467: why are dates preceding reference number?  They are not needed

Answer: They were removed.

  1. Line 467: You mean mycobacteriophages here

Answer: Yes, It was corrected

  1. Line 472: once again this should be mycobacteriophage

Answer: You are right. It was corrected.

  1. Line 472: phages not cells

Answer: Done

  1. Line 473: Really should be citation at end of this sentence.

Answer: In fact, all methods that are going to be explained at the following paragraphs were briefly mentioned in lines between 541-546. To clarify them, they were all cited as follows:

Some of the most-known phage-based diagnostic techniques are as follows: shuttle plasmids [7,19,72,162] and transduction of fluorescent or non-fluorescent foreign DNA into the mycobacterial genome and distinguishing the antibiotic resistance or viability of mycobacteria through fluorescent emission or formation of turbid lysogenic plaques [19,20,163]; phage amplification and detection of the viability of mycobacteria [24,25,164,165]; capture of viable target mycobacteria using mycobacteriophage  proteins [166] or whole phages as ligands [79,87,88].

  1. Line 493: M. bovis

Answer: Done

  1. Line 502: what is meant?

Answer: It was meant automated technologies. Remoted changed to automated.

  1. Line 505: what is meant? sentence doesn't make sense as written

Answer: Line 580-584: the sentence was modified as follows: The auxotrophic strains are usually used as model organisms for the assessment of drug-resistant M. tuberculosis and latent TB strains, in which these mutant strains underwent extensive in-vivo or in-vitro biosafety testing to be qualified for application in biosafety level 2 facilities  [176]

  1. Line 512: Strikethrough “was”

Answer: Done

  1. Line 518: where is figure 2 referred to in the text?

Answer: Please refer to line 578-580.  “Furthermore, in a recent study in 2020, a recombinant TM4 mycobacteriophage containing a nanoluciferase (Nluc) reporter gene cassette was designed and transduced into different viable pathogenic, drug-sensitive and drug-resistant auxotrophic strains of M. tuberculosis (Figure 2) [20]. ” Line 584-590: “Accordingly, drug-susceptibility tests (DST) were carried out, and the performance of transduced TM4-nluc in both viable drug-resistant and drug-sensitive M. tuberculosis strains was evaluated in the presence and absence of antibiotics, and a cellular limit of detection (LOD) of ≤ 102 CFU was suggested for the assay (Figure 2) [20]. Interestingly, this analysis revealed that the expression of nanoluciferase gene in drug-sensitive M. tuberculosis strains was restricted in the presence of some drugs corresponding  with a reduction in generative light signals (Figure 2) [20].”

  1. Line 529: cite in usual scientific forat i.e 6-9x10^5 CFU/mL

Answer: Done

  1. Line 536: was

Answer: It was added

  1. Line 561: mediated add a d at the end

Answer: Done

  1. Figure 3: phage amplification does not happen here. Step 3 is phage amplification step

Answer: Yes, you are right. It was corrected.

  1. Figure 3: biotinylated not biotinylated

Answer: Done

  1. Line 577: not the correct wording. Lawn does not form around lysed plaques. Plaques form around lysed MAP cells in a lawn of M. smegmatis.

Answer: Line 664-667: It was corrected as follows: “Then,  to visualize the viability of MAP cells in samples,  the lysate was cultured with a fast-growing mycobacterium, M. smegmatis on a solid medium, in which plaques that each corresponded to a viable MAP cell or clump of MAP cellsformed around lysed MAP cells in a lawn of M. smegmatis  (Figure 3)[165]’’.

  1. Line 578: cell or clump of MAP cells

Answer: Done

  1. Line 579: Strikethrough “almost”

Answer: Done

  1. Line 586: no new paragraph here

Answer: Done

  1. Line 593: no new paragraph - you are still on the same topic

Answer: Done

  1. Line 595: citation missing here. the new assay was not called phage-bead qPCR, it was termed Phagomagnetic separation-qPCR assay, so change text in figure title.

Answer: It was corrected and cited accordingly. Line 682-684: “In 2020, in an innovative study, mycobacteriophage D29 was directly coupled to magnetic beads via covalent bonds to capture MAP in milk samples through a magnetic separation step [87] (Figure 4). This assay was called phagomagnetic separation-qPCR assay [87]”

Foddai, A.C.G.; Grant, I.R. A Novel One-Day Phage-Based Test for Rapid Detection and Enumeration of Viable Mycobacterium Avium Subsp. Paratuberculosis in Cows’ Milk. Appl. Microbiol. Biotechnol. 2020, 104, 9399–9412, doi:10.1007/s00253-020-10909-0.

  1. Line: No it doesn't. Specificity of the test lies with IS900qPCR step. It doesnt matter if DNA of other mycobacteria is present. Revise this text accordingly

Answer: It was corrected as follows: Line 691-696: Although any viable mycobacterial species that could be infected with D29 might be recovered, and their DNA would be transferred to the lysate, this is the qPCR analysis that influences the specificity of the assay remarkably. Accordingly, in the assessment of the efficiency of three qPCR methods, including SYBR Green qPCR IS900, TaqMan qPCR IS900, and TechneTM PrimePro qPCR DNA detection kit for detection of MAP DNA presented in the same lysate, the diagnostic level considerably improved through TaqMan qPCR IS900 analysis rather than other qPCR methods [87].

  1. Figure 4: size of phages relative to bead is wrong

Answer: The sizes are imaginary. We just wanted to make readers imagine what is going on. To eliminate ambiguity, we mentioned the following sentence in front of the figure title “Note: The sizes are imaginary in this figure, and all components were magnified to make the procedure more understandable”    

  1. Figure 4: similar comment, size of phage is too big relative to mycobacterial cell

Answer: The sizes are imaginary. We just wanted to make readers imagine what is going on. To eliminate ambiguity, we mentioned the following sentence in front of the figure title “Note: The sizes are imaginary in this figure, and all components were magnified to make the procedure more understandable”    

  1. Line 610: Mycobacterium sp. with italics for Mycobacterium

Answer: Done

  1. Line 621: resistant not resistance

Answer: Done

  1. Line 621: not a correct English word. Efficacy or effectiveness would be better words

Answer: It was replaced with effectiveness. 

  1. Line 626: efficacy is correct word

Answer: Done

  1. Line 628: no new paragraph

Answer: Done

  1. Line 633: no new paragraph

Answer: Done

  1. Line 636: Capital letters need for each word in organisation name

Answer: Done

  1. Line 636: existing

Answer: Done

  1. Line 637: Strikethrough “were asked”

Answer: Done

  1. Line 646: ,

Answer: Done

  1. Line 649: what is hilus?

Answer: Hilus or hilum is a depression on the surface of some organs such as spleen, kidney where vessels, nerves, and ducts enter or leave it. This word was taken from the original article

Sula L, Sulová J, S.M. Therapy of Experimental Tuberculosis in Guinea Pigs with Mycobacterial Phages DS-6A, GR-21 T, My-327. Czech Med 1981, 4, 209–214.

  1. Line 652: you need to link these two sentences. Sentence should not start with in which

Answer: Done

  1. Line 665: same comment as above. Sentence shouldn't start with in which

Answer: Done

  1. Line 672: Strikethrough “S”

Answer: Done

  1. Line 692: no new paragraph

Answer: Done

  1. Line 700: no new paragraph - still on same topic?

Answer: Done

  1. Line 709: no new paragraph

Answer: Done

  1. Figure 5: I don't see what value this figure has

Answer: This figure was removed

  1. Line 756: of

Answer: It was added

  1. Figure 6: does this figure add anything to the manuscript?

Answer: The figure was removed

  1. Line 776: seems a strange word. Is there not a more appropriate word to use?

Answer: It was replaced with the word of “damages”

  1. Line 784: double negative in this sentence - replace word none with any

Answer: Corrected

  1. Line: slang expression. what is meant by ditch?

Answer: It was replaced with escape

  1. Line 803: Conclusion section is far too long currently. You need to concisely summarise your findings from the lit review, not start introducing new material

Answer: Conclusion was summarized, and your suggestions and corrections were considered as follows: The exploitation of mycobacteriophages in medicine opened new horizons in diagnosing and treating mycobacterial-associated infections, specifically those caused by M. tuberculosis, M. aviumspp., and M. abscessus. The properties that make mycobacteriophages potential candidates for phage-based diagnosis and therapy against mycobacteria are related to their capability in selective transduction of foreign DNA into mycobacterial genome, formation of superinfection stable lysogens and induction of toxicity against both slow- and fast-growing mycobacteria, production of endolysins (Lysin A, Lysin B, or both) destructive to the mycobacterial cell wall, transportation of small mobile genetic elements,  reduction of the acid-fastness of specific members of mycobacteria, excision of the antimicrobial resistance genes, lysis of non-/-pathogenic mycobacteria in various environments such as the bloodstream and liposomal macrophages. Previous studies depicted that mycobacteriophages in the frame of recombinant shuttle plasmids containing fragments of cloned foreign DNA, antibiotic resistance genes, or nanoluciferase (Nluc) reporter gene cassette could discriminate between drug-sensitive and drug-resistant pathogenic mycobacterial species. In addition, phage amplification-based techniques disclosed essential information about the impact of therapeutic agents on slow-growing mycobacteria (i.e., MAP, M. tuberculosis) and their viability in various sample types such as milk, blood, feces, tissue, etc. Importantly, mycobacteriophages and their expressed proteins could be potential ligands for capture and detection of viable mycobacteria. On the other hand, intact/genetically modified mycobacteriophages and their products, such as lysin, in isolation or within a synergistic antibiotic therapy, demonstrated therapeutic impacts on infections caused by M. tuberculosis and M. abscessus. Mutations induced by mycobacteriophages could also have therapeutic applications, in which mutations in subunits of citrate lyase in M. tuberculosis strains make the bacterium susceptible to oxidative stress and disrupt the normal replication inside macrophages. However, mycobacteria always have different direct or indirect mechanisms to escape infection via mycobacteriophages. One of these strategies that impose struggles on phage therapy is the presence of mycobacteria engulphed in macrophages. Application of non-pathogenic mycobacteria that have already been infected with lytic mycobacteriophages or mycobacteriophages encapsulated in liposome would be promising solutions to this problem, in which they enable mycobacteriophages to cross barriers that exist on the surface of the phagocytic cells and get access to the internalized mycobacteria. Nevertheless, strategies that immunize mycobacteria against infections might fail by concurrent administration of phage cocktails and antibiotics to patients. On the other hand, synergistic therapy might also have unpredicted consequences since patients must take multiple drugs in various quantities and titers of mycobacteriophages. Therefore, careful studies are needed to elucidate the side effects of these combinatory treatments. This should be along with discovering new mycobacteriophages specific to the most struggling pathogenic mycobacteria and finding alternative therapies that have the maximum efficacy and least side effects for patients.

Round 2

Reviewer 2 Report

Thank you for revising your manuscript in accordance with my suggestions and comments on version 1.  I can see that the English and grammar are now much improved, and other corrections have been made to the text. 

I have noted some minor corrections are still needed to the reference section, as follows:

Ref 123 and 124 - author names should be provided, not just initials, for these two citations

Ref 130 - author names provided are first names rather than surnames, so this needs to be corrected. 

Ref 146 - remove capital letters from author names. 

I have also noted irregular line spacing between paragraphs and inconsistent margin sizes in the typeset version of the revised manuscript (may be editorial office issue rather than how you submitted the manuscript?)

Author Response

Please find a point by point response to the reviewer's comments:

Thank you for revising your manuscript in accordance with my suggestions and comments on version 1.  I can see that the English and grammar are now much improved, and other corrections have been made to the text. 

Answer: We are really grateful to the reviewer for the nice comments and for the help in shaping the manuscript.

I have noted some minor corrections are still needed to the reference section, as follows:

Ref 123 and 124 - author names should be provided, not just initials, for these two citations

Answer: Fixed

Ref 130 - author names provided are first names rather than surnames, so this needs to be corrected. 

Answer: Fixed

Ref 146 - remove capital letters from author names. 

Answer: Fixed

I have also noted irregular line spacing between paragraphs and inconsistent margin sizes in the typeset version of the revised manuscript (may be editorial office issue rather than how you submitted the manuscript?)

Answer: We noticed it as well (my be in editorial issue), we fixed all the irregular spacing between paragraphs as well.

Thank again for everything.

Best Regards

Leonardo A. Sechi 

This manuscript is a resubmission of an earlier submission. The following is a list of the peer review reports and author responses from that submission.

Round 1

Reviewer 1 Report

I have thoroughly reviewed this manuscript. Overall, it is a well-written manuscript dealing with an interesting and novel topic. I only have a few minor observations:

  • Please check that these terms should be written in italics:
    • Line 177-178 and 181-182: tuberculosis.
    • Line189: M. smegmatis
  • Please check the correct spelling of the following terms:
    • Line 428: therapiutic
    • Line 429: Transuction
    • Line 669: carful
  • The fourth paragraph in Conclusions is too long. I recommend separating into shorter paragraphs a period.
  • Check all references according to the journal's instructions. For example, some references are in capital letters.

Author Response

Reviewer #1 (Remarks to the Author):

Overview: I have thoroughly reviewed this manuscript. Overall, it is a well-written manuscript dealing with an interesting and novel topic. I only have a few minor observations

Comments: Please check that these terms should be written in italics:

Answer: The terms were checked and formatted to Italic style as you recommended.     

1.Line 177-178 and 181-182: tuberculosis.

Answer: Done

2.Line189: M. smegmatis

Answer: Done

Comment: Please check the correct spelling of the following terms:

3.Line 428: therapeutic

Answer: Done

4.Line 429: Transuction

Answer: Done

5.Line 669: carful

Answer: Done

6.The fourth paragraph in Conclusions is too long. I recommend separating into shorter paragraphs a period.

Answer: It was modified. Please refer to Line 842-865:

The diagnostic properties of various mycobacteriophages against M. tuberculosis, M. avium subsp. paratuberculosis, M. abscessus were disclosed in various studies. Previously, recombinant shuttle plasmids were designed that contained fragments of cloned foreign DNA that could be transduced in genome of fast-growing mycobacteria and express resistance against antibiotics (i.e. Kanamycin) resulting in turbid lysogenic plaques. Also, transformed recombinant plasmids were engineered containing M. fortuitum and E.coli replicons plus antibiotic resistance genes that were injected to both fast- and slow-growing mycobacteria via electroporation, transduced in their genome and expressed there. Interestingly, insertion of fluorescent reporter genes to mycobacteriophages turned them to detective tools that could estimate the susceptibility of mycobacteria to antibiotics/drugs via fluorescent emission. In which, mycobacteriophages (i.e., TM4) containing a nanoluciferase (Nluc) reporter gene cassette could discriminate viable pathogenic/drug-sensitive/drug-resistant strains of M. tuberculosis from each other via light signals.

Furthermore, catabolic reactivity of mycobacterial-induced TOP1A that was extracted by mycobacteriophages was employed in detection of mycobacterial species in crude samples. Phage amplification-based techniques disclosed essential information about the impact of therapeutic agents on slow-growing mycobacteria (i.e., MAP, M. tuberculosis) along with their viability in various samples such as milk, blood, feces, tissue, etc. On the other hand, mycobacteriophages and their expressed proteins could be potential ligands for capturing viable mycobacteria such as MAP, M. tuberculosis, and M. smegmatis in various specimens. This could be followed by molecular analysis of DNA existed in lysate or quantification of other biomarkers such as ATP via emitted bioluminescent signals.

Reviewer 2 Report

This review article by Hosseiniporgham and Sechi reports an update of the currently knowledge on mycobacteriophages. With the emergence of antibiotic-resistant bacterial strains, there has been an increasing interest in phage therapy during the last few years, recently addressed in clinical trials or in compassionate studies, particularly against infections caused by non-tuberculous mycobacteria. This explains the multiplication of recent reviews in the mycobacteriophage field. The present manuscript, however, does not add new data or additional values with respect to the existing reviews on mycobacteriophages. In addition, this manuscript appears very poor in its presentation (only 1 figure) and content. The english is very poor, rendering many sentences unreadable and some section difficult to follow. In addition, some titles do not properly reflect the content of the section.  Clearly, a revised manuscript would require major improvements and corrected by an native-english speaker.

Important points to consider:

-section 2 on morphology is very short. Needs to be amended and illustrated with nice EM pictures.

-section 5 “A brief history about discovered mycobacteriophages” has nothing to do with history. This is just a catalogue of various phages. I propose that the authors write a section dedicated to history and present the phages listed in the text in a Table with appropriate references.

-Figure 1 is poor. More details should be provided. Make this figure more appealing.

-section 8 on phage resistance is interesting but too short. This should be expanded and developed with more details.

-english is poor and should be improved. Here are a few mistakes, among others, that require changes:

-line 12-14, is not correct

-line 27 is not correct

-line 41: “plasmid vectors”, a plasmid is a vector, no need to add both terms

-ine 53, “Myvoviruses”

-line 61: what does 2.5:1 to 4:1 mean ?

-line 137 “M. semegmatis”

-liner 261: “bulletti” instead of “bolletii”

-line 428: “therapiutic”

-line 539: “BP2” are the authors referring to “BPs” ?

-line 607: “gens”

Author Response

Reviewer #2 (Remarks to the Author):

Overview: This review article by Hosseiniporgham and Sechi reports an update of the currently knowledge on mycobacteriophages. With the emergence of antibiotic-resistant bacterial strains, there has been an increasing interest in phage therapy during the last few years, recently addressed in clinical trials or in compassionate studies, particularly against infections caused by non-tuberculous mycobacteria. This explains the multiplication of recent reviews in the mycobacteriophage field. The present manuscript, however, does not add new data or additional values with respect to the existing reviews on mycobacteriophages. In addition, this manuscript appears very poor in its presentation (only 1 figure) and content. The english is very poor, rendering many sentences unreadable and some section difficult to follow. In addition, some titles do not properly reflect the content of the section.  Clearly, a revised manuscript would require major improvements and corrected by an native-english speaker.

Answer: Dear reviewer, we kindly appreciate your helpful comments on the first version of the manuscript. In fact, we believe in the importance of such reviews in enhancement of the overall knowledge of researchers about the latest achievements of other researchers in phage-based analysis. Many papers have been published during the recent years aiming to fulfil this ambition. While they try to restate the background information and most exciting achievements from the past to now, each contains a new message, logic, and novelty that other publications did not have them at all. Accordingly, the present manuscript underwent major revision to improve the contents and English style based on the raised concerns.

Comment: Important points to consider:

1.section 2 on morphology is very short. Needs to be amended and illustrated with nice EM pictures.

Answer: This section underwent modifications and supplemented with new updates and two TEM images

2.section 5 “A brief history about discovered mycobacteriophages” has nothing to do with history. This is just a catalogue of various phages. I propose that the authors write a section dedicated to history and present the phages listed in the text in a Table with appropriate references.

Answer: This section was transferred to after introduction (Line 59) and retitled “A brief history about discovered mycobacteriophages infective to M. smegmatis, M. tuberculosis, M. bovis, M. avium spp., and M. abscessus“.  This part was entirely and carefully revised and redesigned to become more understandable. In new format, bacteriophages of the same function were presented together. Table 1 was added to summarize the most interesting information about phages at the end of section 2 and it was titled “Table 1. Descriptive comparison of some potential mycobacteriophages infective to M. smegmatis, M. tuberculosis, M. bovis, M. avium spp., and M. abscessus”.

3.Figure 1 is poor. More details should be provided. Make this figure more appealing.

Answer: Done (5 figures were added in different section of the manuscript)

Line 754: Figure 6. Two strategies for accessing the pathogenic mycobacteria engulfed in macrophages:  A) Encapsulation of mycobacteriophages of interest in lipoome, B) using non-pathogenic mycobacteria as vessels for transportation of mycobacteriophages to the intracellular spaces of immune cells (i.e., macrophages).

Line 715: Figure 5. Application of mycobacteriophages in isolation/cocktail or within a synergistic antibiotic therapy against mycobacterial infections.

Line 561: Figure 4. The procedure of MAP viability assessment via phage-bead qPCR (MB: Magnetic beads).

Line 515: Figure 3. The procedure of peptide-mediated magnetic separation (PMS) phage assay and detection of viability of MAP in milk samples (MBs: Magnetic beads)

Line 452: Figure 2. Detection of viable drug-sensitive/resistant M. tuberculosis via transducing TM4 containing a nanoluciferase (Nluc) reporter gene cassette into bacterial genome.

Line 339 : Figure 1. TEM images taken from mycobacteriophages TM4 (A) and Rahel (B), members of Siphoviridae (cluster K) [74] and Myoviridae (subcluster C1) [59] families respectively.

Line 306: Table 1. Table 1. Descriptive comparison of some potential mycobacteriophages infective to M. smegmatis, M. tuberculosis, M. bovis, M. avium spp., and M. abscessus  

4.section 8 on phage resistance is interesting but too short. This should be expanded and developed with more details.

Answer: More details were added to this section as recommended.

Line 778-796: Further analysis on the susceptibility of M. abscessus isolates to phage treatment revealed that the morphology of colonies (rough or smooth) extremely influences the resistance of M. abscessus to phage treatment, in which strains that developed rough colonies were more likely (80%) to be infected with and eliminated by at least one mycobacteriophage [186]. However, smooth-colony strains were not responsive to none of selected mycobacteriophages [186]. In fact, the formation of rough colonies in some strains of M. abscessus is induced by insertions and deletions in mps1 and mps2 genes that encode the synthesis of glycopeptidolipides, whereas mutations in polyketide synthesis, uvrD2, and rpoZ genes could induce resistance in strains that produce smooth colonies [186]. Moreover, M. abscessus strains are prone to genetic discrepancy that is directed by prophage and plasmid mobilome. Since prophages and plasmids are existed in a high profusion and variation, and they carry a broad repertoire of genes, these elements could affect the susceptibility, virulence, and defense of M. abscessus strains to mycobacteriophages [187]. In 2021, a study on 82 clinical isolates of M. abscessus demonstrated that majority of the strains (85%) had one or more prophages containing sequences from minimum 17 clusters that were attached to 18 various attB sites. These prophages induced the production of 19 sets of polymorphic toxin and toxin-immunity systems that each marked by WXG-100 proteins which was transported through secretion system type VII [187].     

Comment: English is poor and should be improved. Here are a few mistakes, among others, that require changes:

Answer: The manuscript underwent carful English correction

6.line 12-14, is not correct

Answer: The abstract was entirely modified. Please refer to the manuscript lines 10-25.

7.line 27 is not correct

Answer: It was corrected. Line 36-39: Majority of discovered mycobacteriophages have been isolated using M. smegmatis mc2155 [3].  Some others are only known to infect M. tuberculosis H37Rv [3,4], while others have been isolated using other mycobacterial species.

8.line 41: “plasmid vectors”, a plasmid is a vector, no need to add both terms

Answer: Vector was removed.

9.line 53, “Myvoviruses”

Answer: It was corrected.

10.line 61: what does 2.5:1 to 4:1 mean?

Answer: Units were added accordingly.

Line 330-336: The shape and size of capsid differ by the strains of mycobacteriophages and their genome size respectively. Most mycobacteriophages have isometric capsid that its diameter ranged between 40 to 80 nm, whereas some others such as Corndog, Che9c, and Brujita [134] have prolate heads that dimensioned to length: width ratios of between 2.5:1 to 4:1 [7]. This is under the condition that the tail length and its tip structure [134]impose another variations to the phage structure, to which the length of tail ranges between 105 [43] to 350 nm [7].

11.line 137 “M. semegmatis”

Answer: It was corrected.

12.liner 261: “bulletti” instead of “bolletii”

Answer: Line 293: It should be bolletti based on the reference
103: Sassi, M.; Bebeacua, C.; Drancourt, M.; Cambillau, C. The first structure of a mycobacteriophage, the Mycobacterium abscessus subsp. bolletii phage Araucaria. J. Virol. 2013, 87, 8099–8109, doi:10.1128/JVI.01209-13.

13.line 428: “therapiutic”

Answer: It was corrected.

14.line 539: “BP2” are the authors referring to “BPs” ?

Answer: Yes, it should be BPs and it was corrected accordingly.

15.line 607: “gens”

Answer: It was corrected to genes.

Reviewer 3 Report

This review is on an important topic of mycobacteriophages, where there have been some exciting recent results on phage therapy. The authors are clearly knowledgeable in this area. I found the review very difficult to read however. Hopefully my comments can help the authors for revised versions.

  1. The overall quality of the writing was not good and this review will need major grammatical editing. It was just very hard to read.
  2. Overuse of parentheses in multiples places throughout makes it difficult to read.
  3. There is only 1 Figure. And the legend and figure itself are difficult to understand. Inclusion of several more well-targeted figures and/or tables would greatly help to make it more readable.
  4. The organization is not good. The history of mycobacteriophages doesn't come until section 5. The brief history section also has different paragraphs for all the different types of phages, but it reads rather like an atlas or index (very descriptive without development of interesting concepts), and the phages are not grouped together into common functional themes that would make it more interesting.
  5. In many places it is difficult to decipher what the latest exciting developments within the last ~5 years were. In reading I think something is "recent", only to find a reference for it from the 1990s.
  6. Phage therapy is one of the more exciting developments. I didn't get a sense of what the major challenges were in making this more broadly successful.

Author Response

Reviewer #3 (Remarks to the Author):

Overview: This review is on an important topic of mycobacteriophages, where there have been some exciting recent results on phage therapy. The authors are clearly knowledgeable in this area. I found the review very difficult to read however

Answer: The manuscript underwent a carful and extensive revision to relieve the raised concerns.

Comment: Hopefully my comments can help the authors for revised versions.

1.The overall quality of the writing was not good and this review will need major grammatical editing. It was just very hard to read.

Answer: Proofreading was carried out to diminish grammatical, dictation, and style problems as much as possible.

2.Overuse of parentheses in multiples places throughout makes it difficult to read.

Answer: The number of parentheses reduced to minimum as you recommended.   

3.There is only 1 Figure. And the legend and figure itself are difficult to understand. Inclusion of several more well-targeted figures and/or tables would greatly help to make it more readable.

Answer: One table (Table 1), two TEM images, and 5 figures were added:

5 figures were added in different section of the manuscript)

Line 754: Figure 6. Two strategies for accessing the pathogenic mycobacteria engulfed in macrophages:  A) Encapsulation of mycobacteriophages of interest in lipoome, B) using non-pathogenic mycobacteria as vessels for transportation of mycobacteriophages to the intracellular spaces of immune cells (i.e., macrophages).

Line 715: Figure 5. Application of mycobacteriophages in isolation/cocktail or within a synergistic antibiotic therapy against mycobacterial infections.

Line 561: Figure 4. The procedure of MAP viability assessment via phage-bead qPCR (MB: Magnetic beads).

Line 515: Figure 3. The procedure of peptide-mediated magnetic separation (PMS) phage assay and detection of viability of MAP in milk samples (MBs: Magnetic beads)

Line 452: Figure 2. Detection of viable drug-sensitive/resistant M. tuberculosis via transducing TM4 containing a nanoluciferase (Nluc) reporter gene cassette into bacterial genome.

Line 339 : Figure 1. TEM images taken from mycobacteriophages TM4 (A) and Rahel (B), members of Siphoviridae (cluster K) [74] and Myoviridae (subcluster C1) [59] families respectively.

Line 306: Table 1. Table 1. Descriptive comparison of some potential mycobacteriophages infective to M. smegmatis, M. tuberculosis, M. bovis, M. avium spp., and M. abscessus  

4.The organization is not good. The history of mycobacteriophages doesn't come until section 5. The brief history section also has different paragraphs for all the different types of phages, but it reads rather like an atlas or index (very descriptive without development of interesting concepts), and the phages are not grouped together into common functional themes that would make it more interesting.

Answer: You are correct. This section was moved to after introduction (Line 59) and retitled “A brief history about discovered mycobacteriophages infective to M. smegmatis, M. tuberculosis, M. bovis, M. avium spp., and M. abscessus“.  This part was entirely and carefully revised and redesigned to become more understandable. In new designation, bacteriophages of the same function were presented together. Table 1 was added to summarize the most interesting information about phages at the end of section 2 and it was titled “Table 1. Descriptive comparison of some potential mycobacteriophages infective to M. smegmatis, M. tuberculosis, M. bovis, M. avium spp., and M. abscessus”

5.In many places it is difficult to decipher what the latest exciting developments within the last ~5 years were. In reading I think something is "recent", only to find a reference for it from the 1990s.

Answer: to reduce any ambiguities in classification of data, the contents in phage diagnosis and phage therapy were rearranged from the oldest to latest.

Line 378-575: Mycobacteriophages and detection of pathogenic mycobacteria  

Line 577-768: Mycobacteriophages and treatment of mycobacterial infections

6.Phage therapy is one of the more exciting developments. I didn't get a sense of what the major challenges were in making this more broadly successful

Answer: The following paragraphs were dedicated to address the major reasons that have made phage therapy as a potential approach underlying the major challenges that phage-based treatment might face if it is supposed to do more than a clinical trial and be applied in larger scale of patients. These include the necessity for isolating more phages; development of efficient screening techniques that quickly distinguish the therapeutic phages from others; development of efficient phage-based treatment strategies that can be effective against biofilms; establishment of a safe and certified phage preparation step that play a critical role in large-scale production and formulation of phages;  providing the condition that can guarantee the stability of phages during storage and transportation.  

Line 577-594: Mycobacterial-associated infections have remained a serious concern for decades [166]. Executing screening measures along with appropriate antibiotic therapies play critical roles in treatment of these infections [3]. However, emergence of multi-drug resistance (MDR) mycobacterial species undermined the effectuality of the current antibiotics/drugs remarkably [167].

Nowadays, infections caused by M. tuberculosis, M. avium, and M. abscessus [168] exposed humans, specifically immunodeficient patients (i.e., AIDS), to life-threatening conditions. As an example, in 1998, the administration of protease inhibitors for treatment of infections caused by human immunodeficiency virus type 1 (HIV-1) inhibited M. avium bacteremia in these patients [169]. However, anti-HIV-1 drugs lost their efficiency against MAP infection in these patients soon after development of resistance to the drugs [170].

Phage therapy has opened new horizon in treatment of mycobacterial infections that are insensitive to antibiotic therapy. Intact/genetically modified mycobacteriophages and their products such as lysin, in isolation or within a synergistic antibiotic therapy, influence the life cycle of mycobacteria through either lysing the host cells or inducing mutations to mycobacterial genomes and that would be followed by disruption of mycobacterial replication in host cells.

Line 759-768: Beside the advantages of phage therapies in improvement of the quality and velocity of treatment against mycobacterial infections, some measurements should be taken into account in confronting unpredicted challenges. In fact, major challenges in phage therapy would appear when it is applied in scales larger than clinical trials. Under this circumstance phage therapy relies on isolating more phages; development of efficient screening techniques that quickly distinguish the therapeutic phages from others; development of efficient phage-based treatment strategies that can be effective against biofilms; establishment of a safe and certified phage preparation step that play a critical role in large-scale production and formulation of phages; providing the condition that guarantees the stability of phages during storage and transportation [183]

Reviewer 4 Report

I have uploaded a document that describes my review.

Author Response

Reviewer #4 (Remarks to the Author):

Overview: An update on Mycobacteriophages; From classification to Applications is a problematic read because the use of the English language is poorly done. The specific lines provide many, but not all, examples of this. I also want to say that this manuscript appears to be written by 2 distinct authors, one that used the literature to capture the diversity of the mycobacteriophages over time. In general, this entire section 1 -272 is not deem, by this reviewer, to be worthy of publication. However, the remaining sections have a better command of English and convey the literature in a more comprehensive light. In particular, when the author wrote about their own work, it was more easily comprehended. In addition, as stated in Line 74 of the line items, the authors used data from phagesDB and did not provide the requested acknowledgement for use of this data. Please go back to phagesDB and ask permission to use the data

Answer: Dear reviewer, we really appreciate your helpful comments that motivated us to improve the manuscript. Accordingly, the manuscript underwent extensive revision and proofread to reduce the grammatical and verbal problems to minimum.

Comments:

  1. Line 1-272: is not deem, by this reviewer, to be worthy of publication.

Answer: This part includes abstract, introduction, morphology, classification, life cycle of mycobacteriophages, and a brief history about discovered mycobacteriophages. This section was reviewed and some modifications were implemented accordingly. To increase the readability of and arrangement of the manuscript, section 5 “brief history about discovered mycobacteriophages’’ was moved to after introduction (Line 65) and retitled “A brief history about discovered mycobacteriophages infective to M. smegmatis, M. tuberculosis, M. bovis, M. avium spp., and M. abscessus“.  This part was entirely and carefully revised and redesigned to become more understandable. In new designation, bacteriophages of the same function were presented together. Table 1 was added to summarize the most interesting information about phages at the end of section 2 (table title: “Table 1. Descriptive comparison of some potential mycobacteriophages infective to M. smegmatis, M. tuberculosis, M. bovis, M. avium spp., and M. abscessus”

To find the details, please refer to the manuscript line 10 to 377.

  1. 2. In addition, as stated in Line 74 of the line items, the authors used data from phagesDB and did not provide the requested acknowledgement for use of this data. Please go back to phagesDB and ask permission to use the data

Answer: Thanks for the reminding. You are right, we used some unpublished data taken from PhageDB website. We communicated with PhageDB website and received their consent on publishing the data. I can send the context of communication to the Editor via email on request. In addition, we acknowledged the website at the end of manuscript.

  1. Line 8-9 Use of the verb ‘could’ is odd. I would remove it.

Answer: It was removed as you recommended.

  1. Line 12-13 incomplete sentence Since mycobacterial infections are mostly difficult-to-recognize and insensitive-to-treat due to either slow-growth characteristic of these mycobacteria or their resistance to the common antibiotic therapies.

Answer: This section was modified as follows: Line 10-16

Mycobacterial infections are defined as a group of life-threatening conditions that are triggered by fast- or slow-growing mycobacteria. Some mycobacteria such as M. tuberculosis promote the deaths of million lives throughout the world annually. Although, some nontuberculosis mycobacteria are still responsive to few antibiotics, majority of mycobacterium tuberculsis (Mtb) infections are insensitive to common antibiotic treatments and this is due to the emergence of multi-drug resistance in this bacteria.  On the other hand, detection of mycobacterial infections is always demanding due to the intracellular nature of these pathogens, which along with lipid-enriched structure of cell wall makes the access to the internal contents of cells difficult.

  1. Line 15 -16 Again, use of language here is a problem. What about phages for treatment of Mtb? That is just as desirable.

Answer: This part was modified as follows: Line 19-25:  Although, the infectivity of majority of discovered mycobacteriophages has been evaluated in non-pathogenic M. smegmatis, more researches are still ongoing to discover mycobacteriophages specific to pathogenic mycobacteria such as DS6A that its infectivity has been unveiled against members of M. tuberculosis complex. Accordingly, this review aimed to introduce some of the potential mycobacteriophages in research specifically those which are infective to the three troublesome mycobacteria, Mycobacterium tuberculosis, Mycobacterium avium subsp. paratuberculosis, and Mycobacterium abscessus, highlighting their theranostics applications in medicine.

  1. Line 17. In correct use of ‘exploits’.

Answer: It was changed. Line 22-25: Accordingly, this review aimed to introduce some of the potential mycobacteriophages in research specifically those, which are infective to the three troublesome mycobacteria, Mycobacterium tuberculosis, Mycobacterium avium subsp. paratuberculosis, and Mycobacterium abscessus,highlighting their theranostics applications in medicine.

  1. Line 27: these numbers are out-of-date. The reference cited doesn’t make sense Line 27 e in the context of the sentence.

Answer: It was updated according to the content of PhageDB website and the last review of Prof. Hatfull “Actinobacteriophages: Genomics, Dynamics, and Applications”, published in 2020.  

Line 35-37: Members of these two families are distinguished based on their morphological and evolutional divergences, in which they are distributed into 32 clusters [1] and 10 singletons [2] to date.

  1. Line 27-28. Poor sentence structure. Better to say something like, Most myocbacteriophages found have been isolated using M. smegmatis mc2155. Some are only known to infect M.tuberculosis, while others have been found using other mycobacterial strains. Host range studies are not complete, but – this is where one could then interpret the data to suggest how broad some phage host range is or isn’t.

Answer: You are right. There are many mycobacteriphages that have not been sequenced yet. Accordingly the sentence was changed as follows: Line 37-39: Majority of discovered mycobacteriophages have been isolated using M. smegmatis mc2155 [3].  Some others are only known to infect M. tuberculosis H37Rv [3,4], while others have been isolated using other mycobacterial species

  1. Line 36: some, if not most Mtb infections DO respond to antibiotics. Please revise.

Answer: Done

  1. Line 52; this line does not make sense to me: Siphoviridae and Myoviridae families have, respectively, 61-70 and 9 geniuses that are distinguished by differences that introduced in their genome and caudal structures.

Answer: The sentence has been rewritten as follows: Line 314-318: However, in the latest classification, mycobacteriophages were stratified into two main families of Siphoviridae and Myoviridae including 11/210 and 5/87 subfamily/genera respectively[131]. Members of these families are distinguished by morphological differences such as tail structure [131].

  1. Line 59: the use of ‘in which’ is problematic. The sentence could begin with Most….

Answer: It was corrected.

  1. Lines 67 -69. The authors of the clustering designation of mycobacteriophages do not agree with the statement of 50%. It is much more nuanced that that.

Answer:  It was corrected. Line 343-346: The members of a same cluster have a genomic uniformity of above 35%, whereas the similarity of less than 35% would place a mycobacteriophage to another cluster [135].

  1. Line 73 – I believe that 1655 is incorrect.

Answer: It was corrected.

  1. Line 74- author’s refer to obtaining data from phagesDB.org. However no acknowledgement is provided for the use of this information. Terms of use are found here.

Answer: Thanks for the reminding. We contacted the website and received their consents on right of publishing this data.

  1. Line 116: I don’t think that the tail length differences are real

Answer: This part was removed.

  1. Line 117: genius should likely be genus

Answer: Done

  1. Line 120: ‘believed’ is not an appropriate word choice

Answer: It changed to reported. Line 90

  1. Line 155: attitude is the wrong word choice, attribute may be better

Answer: Done.

  1. Line 150- 155. Refers to an ape gen, that in the title of the cited article is called the aph gene

Answer: Yes, you are right. It should be aph.

  1. Line 157-166. There is additional work done by M Piuri. https://pubmed.ncbi.nlm.nih.gov/30570720/

Answer: This work was included in the manuscript.

Line 427-433: In 2019, mycobacteriophages have been engineered containing fluorescent reporter genes of gfp, ZsYellow, and mCherry that could estimate the susceptibility of mycobacteria against antibiotics/drugs through emission of fluorescent lights in the frame of a fluorescence microscopy, flow cytometry, multiwell fluorimeter analysis. This innovation could facilitate the process of detection of multidrug resistant mycobacteria such as Mycobacterium tuberculosis that normally requires time, remoted-technologies, and abundant finance [151].       

  1. Line 176-181. This work is predated by work with Mycobacteriophage BPs. Sampson T, Broussard GW, Marinelli LJ, Jacobs-Sera D, Ray M, Ko CC, Russell D, Hendrix RW, Hatfull GF. Mycobacteriophages BPs, Angel and Halo: comparative genomics reveals a novel class of ultrasmall mobile genetic elements. Microbiology (Reading). 2009 Sep;155(Pt 9):2962-2977. doi: 10.1099/mic.0.030486-0. Epub 2009 Jun 25. PMID: 19556295; PMCID: PMC2833263.

Answer: This work was included in the manuscript.

Line 144-158: Interestingly, three mycobacteriophages of BPs, Angel, and Halo have the smallest genome size among other mycobacteriophages with 41901 bp, 42289 bp, and 41441 bp respectively [55]. Analysis depicted that BPs and Halo could also replicate in M. tuberculosis less effectively producing the lower number of plaques on plates. Further studies depicted that the capability of Halo in producing the lysed plaques in a lawn of M. tuberculosis has increased through replating a single lysed plaques taken from the primary M.tuberculosis lawn. In fact, the ability of Halo and BPs in formation of lower number of plaques in original plate containing M. tuberculosis lawn came from the inefficiency of phages in identification of their receptors on M. tuberculosis cells [55]. This suggests that replating Halo in new M. tuberculosis lawn would induce new mutations to the phage genome and this may enhance the chance of competent attachment of phage to M. tuberculosis receptors significantly. Despite the importance of all three phages, more researches have been focused on BPs and Angel due to the presence of an insertion of two extremely small mobile genetic elements, MPME1 and MPME2 respectively. These elements have uniform genomic structure, however analysis on pre- and post-integration sequences unveiled that a particular 6 bp insertion existed at one end of each element. These characteristics plus the presence of unusual lysogeny modules in BPs, Angel, and Halo with an attachment attP site that is embedded in repressor site could cut down the repressor gene by each phage integration. These factors propose the three mycobacteriophages as potent therapeutic tools in the future [55].

  1. 193-194: I would not call smeg and Tb “Its hosts species”, rather it was found to infect these.

Answer: Done. Line 238-2406: CRB2 is a typical lytic mycobacteriophage.  Primary study on isolation of CRB2 depicted that it could effectively infect both M. smegmatis and M. tuberculosis [77].

  1. Line 198: do not use recent, but rather include when it was found

Answer: This part underwent modification. Please refer to Line 159-164

Faze9 [56] and Donny [50] are members of cluster B mycobacteriophages and KingMidas is a genus of cluster F [57]. Up to now, 202 mycobacteriophages have been placed within cluster F [1]. All these three phages lack transfer-messenger RNA (tmRNA) or tRNA genes in their genomes [50,56,57]. The recent feature is common among cluster B phages [58]. Genomic analysis demonstrated that Faze9 genome contains various genes involving in DNA packaging, DNA replication (RuvC-like resolvase, DNA helicase, DNA primase/polymerse), host lysis (lysin A and holin), and DNA modifications [56].

  1. Lines 130 -241: sometimes you choose to include cluster designations and sometimes not. Some phages included have not been completely sequenced. Since this is a review article, the completeness of the dataset is rather a key point.

Answer: I agree with you statement. In order to retain the consistency of context and increase the readability of this section, the section was totally rewritten and rearranged based on the application/functionality of mycobacteriophages and the species of mycobacteria that was used for isolation of mycobacteriophages. This section was moved to after introduction. Please refer to Line 65-304. In addition, a table was added at the end of section 2 containing useful and brief information about mhycobacteriophages that were discussed throughout the context.

  1. Lines 205-207: I don’t understand what the value of similarity to Scottish means. There are currently 202 Cluster F phages, so addressing something meaningful would be good.

Answer: The following sentences were added to the manuscript. Line 159-164: Faze9 [56] and Donny [50] are members of cluster B mycobacteriophages and KingMidas is a genus of cluster F [57]. Up to now, 202 mycobacteriophages have been placed within cluster F [1]. All these three phages lack transfer-messenger RNA (tmRNA) or tRNA genes in their genomes [50,56,57]. The recent feature is common among cluster B phages [58]. Genomic analysis demonstrated that Faze9 genome contains various genes involving in DNA packaging, DNA replication (RuvC-like resolvase, DNA helicase, DNA primase/polymerse), host lysis (lysin A and holin), and DNA modifications [56]. Donny genome consists of genes that encode structural proteins, terminase, and Lysin A and B proteins [50].

Line 165-169: KingMidas contains 105 protein-coding genes in its genome [57]. Additionally, typical virion structural/assembly genes were recognized in the mycobacteriophage genome that induce the production of small/large subunits of terminase, portal protein, capsid maturation protease, scaffolding protein, major capsid protein, head-to-tail stopper, tail terminator, major tail protein, tail assembly chaperones, tape measure protein, and minor tail proteins [57].  

  1. Line 208-209: this is again a poor choice of phages to represent Cluster C1. A key component of Cluster C1 phages are that they are generalized transducers.
  2. Line 221-223. There are many phages found in China previously. An example is Ch9c.

Answer: SWU1 data was supplemented with other useful information. However, information related to your recommended phage Ch9C was not found in any publications. Instead of that some information about Ch12C was included to the manuscript.

Line 135-137: Interestingly, a study on mycobateriophage Che12 revealed that its Gp11 has Chitinase domains that functions as Lysin A. Che12 lysin A cuts off NAG-NAM-NAG in peptidoglycan structure in mycobacterial cell wall and produce tautomers of NAG-NAM-NAG [54].

Line 2619-267: Che12 has been recognized as the first temperate mycobacteriophage that could infect and lysogenize M. tuberculosis [82]. Che12 has genome similarity of above 80% with L5 and D29, in which phylogenic studies suggested that Che12 might be derived from L5. Che12 has 70 ORFs containing more than 30 codons [82]. Che12 lysogeny is promoted by integrase, excisionase, and repressor protein. The putative integrase that facilitates the recombination of Che12 to specific site in lysogeny is encoded by ORF27. The phage attachment site attP has homology to attB in M. smegmatis and M. tuberculosis and this capability could be exploited in phage therapy against M. tuberculosis[82].  

  1. Line 229-210: An ‘unusual lysogenic pattern is an incorrect way to say that the plaque morphology is unqiue. But is it?

Answer: I agree with your statement. The sentence was corrected. Line 223-225: TM4 belongs to subcluster K2 [61,63,73]. In contrast to other members of cluster K that can form stable lysogens and turbid plaques, TM4 has  a unique plaque morphology and  creates hazy [63] to clear plaques in solid media [73,74].

  1. Line245: TM4 does not have gp45.

Answer: It was corrected. Line 189-190: This gene that is encoded by prophages also confers immunity to Milly and Adephagia but not TM4, since gp45 has been deleted in TM4 [61].

  1. Line 247-248: More specifically. M. smegmatis mc2155

Answer: It was added.

  1. Line 267-272: the description of M. abscessus prophages is missing the work of Dedrick, et al 2021 (2 papers) In the areas of this paper,

Answer: These works were included in the section 8 (Phage resistance) of the manuscript because they had contents (such as morphology of colonies in M. abscessus and the genetic discrepancy in M. abscessus that is directed by prophage and plasmid mobilome) related to phage resistance that should be explained along with other factors that promote resistance against mycobactreiophages in mycobacteria.

Line 778-796: Further analysis on the susceptibility of M. abscessus isolates to phage treatment revealed that the morphology of colonies (rough or smooth) extremely influences the resistance of M. abscessus to phage treatment, in which strains that developed rough colonies were more likely (80%) to be infected with and eliminated by at least one mycobacteriophage [186]. However, smooth-colony strains were not responsive to none of selected mycobacteriophages [186]. In fact, the formation of rough colonies in some strains of M. abscessus is induced by insertions and deletions in mps1 and mps2 genes that encode the synthesis of glycopeptidolipides, whereas mutations in polyketide synthesis, uvrD2, and rpoZ genes could induce resistance in strains that produce smooth colonies [186]. Moreover, M. abscessus strains are prone to genetic discrepancy that is directed by prophage and plasmid mobilome. Since prophages and plasmids are existed in a high profusion and variation, and they carry a broad repertoire of genes, these elements could affect the susceptibility, virulence, and defense of M. abscessus strains to mycobacteriophages [187]. In 2021, a study on 82 clinical isolates of M. abscessus demonstrated that majority of the strains (85%) had one or more prophages containing sequences from minimum 17 clusters that were attached to 18 various attB sites. These prophages induced the production of 19 sets of polymorphic toxin and toxin-immunity systems that each marked by WXG-100 proteins which was transported through secretion system type VII [187].     

  1. namely Line 388- 438 where this author describes his own work, it is well written. Almost to the extent, that the previous review of literature and this experimental work does not quite fit together.

This work was fitted to the previous literatures as follows: Line 531-575:

On the other hand, mycobacteriophages and their expressed proteins can efficiently capture viable mycobacteria such as MAP, M. tuberculosis, and M. smegmatis in various specimens.

In 2014, an analysis on mycobacteriophage L5 revealed that the genome of this phage encodes proteins that could be utilized as potential ligands in capture of viable MAP and M. smegmatis [165]. In which, immobilized tail protein (Gp6) could capture both MAP and M. smegmatis in samples, whereas lysine protein (Gp10) binds more specifically to M. smegmatis [165]. Experiments on specificity of Gp6 and Gp10 demonstrated that these proteins could specifically target MAP and M. smegmatis neither other mycobacterial species such as Mycobacterium marinum nor Gram-negative bacteria such as E. coli, salmonella, and campylobacter. However, Gp6 could also bind to some of the chemically-synthetized superficial mycobacterial glycans that may undermine its specificity [165].   

In 2020, in an innovative study, mycobacteriophage D29 was directly coupled to magnetic beads via covalent bonds in order to capture MAP in milk samples through magnetic separation step (Figure 4). In another level, recovered viable MAP cells were resuspeneded in Middlebrook (MB) 7H9 supplemented with 10% Oleic Albumin Dextrose Catalase (OADC) and 2 mM CaCl2 and that was followed by the incubation of suspensions at 37 ºC/ 2 hours and a thermal shock at 55 ºC/ 1[87]-2 min [88]. Later, the presence of MAP DNA in lysates (Figure 4) was assessed by qPCR IS900 analysis [87]. The modified phage assay introduced a meaningful sensitivity along with velocity to the procedure of MAP viability assessment, in which the length of diagnosis reduced from 48 hours (PMS-phage assay) to almost 7 hours with a LOD of 10 MAP cells per 50 mL [87] or 10 mL [88] milk. However, this method had a lower specificity than PMS-phage assay, since any viable mycobacterial species that could be infected with D29 might be recovered and their DNA would be transferred to lysate.  In 2020 and 2021, this method  was tested on bovine[87] and sheep/goat [88] milk samples and it sensitively detected viable MAP in 49% out of 100 and 48.78% out of 41 of the studied animals respectively [87,88].

In 2021, mycobacteriophage SWU1 was similarly used as ligands for coating magnetic beads in order to retrieve viable M. smegmatis in samples. M. smegmatis was selected as a model mycobacterium in this analysis, since it is fast-growing and it resembles the pathogenic mycobacteria (i.e., M. tuberculosis) in terms of physiological characteristics [80]. Therefore, the numbers of viable M. smegmatis cells in samples were estimated through quantification of bioluminescent signals emitted by intracellular adenosine triphosphate (ATP) during lysis (after 60 min replication) of viable cells that were already captured and infected with SWU1 [80]. This method effectively detected viable M. smegmatis in various human specimens such as saliva, urine, and serum at minimum concentration of 3.8 × 102 CFU mL-1 (this point was adjusted as the limit of detection (LOD) of the assay) [80].

  1. 658- bacteriophages are neither pathogenic or non-pathogenic

Answer: You are right. It was corrected

  1. 662- bacteriophages are not lysogenic (but rather temperate)

Answer: it was corrected

  1. 672: maybe the author meant reveal instead of relive.

Answer: it was corrected and changed to relieve.

Round 2

Reviewer 2 Report

The authors have considerably improved the quality of the manuscript and have responded to my be previous concerns. They have largely increased the number of figures/tables to illustrate the content of the review, making it much easier to follow.

Reviewer 4 Report

Once again, I find many parts of this manuscript problematic. 

Review for:  An update on Mycobacteriophages; From classification to Applications

An update on Mycobacteriophages; From classification to Applications is improved from the first time.  I will again re-iterate that the second half of the manuscript is in better shape the review parts of this manuscript.

Here are some of the specifics from the 2nd review:

I am pleased to see some of the corrections offered by this author.  The use of English has improved.

In addition, obtaining permission to use data from phagesDB was expected and accomplished for data.  Specifically, though, Figure 1 are electron micrographs of phages TM4 and Rahel from phagesDB.  And no permission was requested or granted.

12 and 44 -45 :I do not believe that most TB strains are resistant to antibiotics.  My quick search said 20% to one antibiotic, so that is not most.  However, when TB is resistant to all antibiotics, an alternative is needed.

17:  accelerated is misspelled

134-135, 138, 139, 151….:  be consistent with lysin A, lysin B vs Lyse A, Lyse B.

145-146:  just because a holin was not computationally found, does not mean that phage had no holin.  I believe current thought it that in order for a cell to lysis from phage infection, a holin – or the activity of a holin – is essential.  A quick transmembrane scan of the described genome reveals 11 different proteins with transmembrane domains detected.  Were they investigated?

1324 please check the reference (151).  It appears the author list is amiss.

173- 179:  I think that this is an incorrect interpretation of the data presented in this article.  I  think that the was in duction of a mutant, but rather a selection for a mutation.

188-189:  Again, I am not sure I understand this interpretation “could cut down the repressor gene by each phage integration.  And then why would this be true? “These factors propose the three mycobacteriophages as potent therapeutic tools in the future.”

215-217:  the whole issue of why phages have tRNAs is not clear.  This paragraph captures the meaning behind the reference used, BUT I am still not sure of its value to an article that is a review of the mycobacterium.  This issue would fall off my top 100 big ideas of the mycobacteriophages.  What does this ; “This is because many phages existed in close clusters can induce lysis of mycobacterial hosts regardless of the presence or absence of tRNA genes in phages genomes.” Mean?

268:  I find the use of the word unique (as in other places in this manuscript) to be problematic.   Shouldn’t a review article be able to describe in what capacity something is unique?

Table 1:  It is odd to me that many phages listed in Table 1 have but a brief mention in a Microbial Resource Announcement – with only a 500 word cut-off of which most has to describe the sequencing information - while the mycobacteriophages that have been studied in depth have been overlooked.  This is especially true as so many phages found in this literature review have not been sequenced.

378:  Table 1 and a simple Blastn would easily show that phiT46-1 belongs to cluster MabG and is like prophages found in M. abscessus strains, Dedrick et al 2021.

277:  I don’t believe that TM$ has a unique plaque morphology.

310:  again, unique plaque morphology for SWU1 is not true.  And plaques do not lyse.

In general, I would be cautious about how one uses plaque morphologies to distinguish amongst phages.  Plaque morphology is keenly dependent on environmental factors that include type of media, type of reagents in the media, time, temperature. 

325:  I don’t think this is true, L5, published in Pedulla, et al 2003 shows its integration cassette.

491-493:  I am not sure why this author keeps saying this.  That somehow mycolic acids are more difficult to adsorb to is not substantiated.  Mycobacteriophages infect nycobacteria.  While we may not understand it well, doesn’t mean the phages are having a nick of trouble.

Figure 1:  What I the source of these photos?

422-423:  while this sentence is now true, it misses a bigger point that not all clusters are alike.  Some are quite homogenous, and their genetic similarity is at 98% while others do indeed only show 35% similarity.  Clusters are merely used to pragmatically speak of this vast collection of phages.  In other words, I would dissuade this author from using the word classify.  Classification is a scientific word that has a vast meaning that cannot be applied to phages.  See Lawrence et al, 2002.

453- Up until now?  I don’t understand.  The literature is 20 years old.

462-466:  Most of the 2120 mycobacteriophages listed on phagesDB have a listing for their lifestyle.  Odd that this author makes it sound so elusive.

483:  I think you mean mycobacteriophages

Section  6:  The omission of 3 mycobacteriophage engineered tools – BRED, CRISPY-BRED and CRISPY-BRIP is huge.  These tools have revolutionized the ability to engineer phages for genetic studies, diagnostic tools and therapeutic treatment.
